# ESTIMATING LIPSCHITZ CONSTANTS OF MONOTONE DEEP EQUILIBRIUM MODELS

**Chirag Pabbaraju**[*] **& Ezra Winston**[*]
Carnegie Mellon University
{cpabbara, ewinston}@cs.cmu.edu

**J. Zico Kolter**
Carnegie Mellon University
Bosch Center for Artificial Intelligence
zkolter@cs.cmu.edu

## ABSTRACT

Several methods have been proposed in recent years to provide bounds on the Lipschitz constants of deep networks, which can be used to provide robustness guarantees, generalization bounds, and characterize the smoothness of decision boundaries. However, existing bounds get substantially weaker with increasing depth of the network, which makes it unclear how to apply such bounds to recently proposed models such as the deep equilibrium (DEQ) model, which can be viewed as representing an infinitely-deep network. In this paper, we show that monotone DEQs, a recently-proposed subclass of DEQs, have Lipschitz constants that can be bounded as a simple function of the strong monotonicity parameter of the network. We derive simple-yet-tight bounds on both the input-output mapping and the weight-output mapping defined by these networks, and demonstrate that they are small relative to those for comparable standard DNNs. We show that one can use these bounds to design monotone DEQ models, even with e.g. multiscale convolutional structure, that still have constraints on the Lipschitz constant. We also highlight how to use these bounds to develop PAC-Bayes generalization bounds that do not depend on any depth of the network, and which avoid the exponential depth-dependence of comparable DNN bounds.

## 1 INTRODUCTION

Measuring the sensitivity of deep neural networks (DNNs) to changes in their inputs or weights is important in a wide range of applications. A standard way of measuring the sensitivity of a function $f$ is the *Lipschitz constant* of $f$, the smallest constant $L \in \mathbb{R}_+$ such that $\|f(x) - f(y)\|_2 \leq L\|x - y\|_2$ for all inputs $x$ and $y$. While exact computation of the Lipschitz constant of DNNs is NP-hard (Virmaux & Scaman, 2018), bounds or estimates can be used to certify a network's robustness to adversarial input perturbations (Weng et al., 2018), encourage robustness during training (Tsuzuku et al., 2018), or as a complexity measure of the DNN (Bartlett et al., 2017), among other applications. An analogous Lipschitz constant that bounds the sensitivity of $f$ to changes in its weights can be used to derive generalization bounds for DNNs (Neyshabur et al., 2018). A growing number of methods for computing bounds on the Lipschitz constant of DNNs have been proposed in recent works, primarily based on semidefinite programs (Fazlyab et al., 2019; Raghunathan et al., 2018) or polynomial programs (Latorre et al., 2019). However, as the depth of the network increases, these bounds become either very loose or prohibitively expensive to compute. Additionally, they are typically not applicable to structured DNNs such as convolutional networks which are common in everyday use.

The deep equilibrium model (DEQ) (Bai et al., 2019) is an *implicit-depth* model which directly solves for the fixed point of an "infinitely-deep", weight-tied network. DEQs have been shown to perform as well as DNNs in domains such as computer vision (Bai et al., 2020) and sequence modelling (Bai et al., 2019), while avoiding the large memory footprint required by DNN training in order to backpropagate through a long computation chain. Given that DEQs represent infinite-depth networks, however, their Lipschitz constants clearly cannot be bounded by existing methods, which are very loose even on networks of depth 10 or less.

---

[*]Equal contribution.

In this paper we take up the question of how to bound the Lipschitz constant of DEQs. In particular, we focus on *monotone* DEQs (monDEQ) (Winston & Kolter, 2020), a recently-proposed class of DEQs which parameterizes the DEQ model in a way that guarantees existence of a unique fixed-point, which can be computed efficiently as the solution to a monotone operator splitting problem. We show that monDEQs, despite representing infinite-depth networks, have Lipschitz constants which can be bounded by a simple function of the strong-monotonicity parameter, the choice of which therefore directly influences the bound. We also derive a bound on the Lipschtiz constant w.r.t. the weights of the monDEQ, with which we derive a deterministic PAC-Bayes generalization bound for the monDEQ by adapting the technique of (Neyshabur et al., 2018). While such generalization bounds for DNNs are plagued by exponential dependence on network depth, the corresponding monDEQ bound does not involve any depth-like term.

Empirically, we demonstrate that our Lipschitz bounds on fully-connected monDEQs trained on MNIST are small relative to comparable DNNs, even for DNNs of depth only 4. We show a similar trend on single- and multi-convolutional monDEQs as compared to the bounds on traditional CNNs computed by AutoLip and SeqLip (Virmaux & Scaman, 2018), the only existing methods for (even approximately) bounding CNN Lipshitz constants. Further, our monDEQ generalization bounds are comparable with bounds on DNNs of around depth 5, and avoid the exponential dependence on depth of those bounds. Finally, we also validate the significance of the small Lipschitz bounds for monDEQs by empirically demonstrating strong adversarial robustness on MNIST and CIFAR-10.

## 2 BACKGROUND AND RELATED WORK

**Lipschitz constants of DNNs** Lipschitz constants of DNNs were proposed as early as Szegedy et al. (2014) as a potential means of controlling adversarial robustness. The bound proposed in that work was the product of the spectral norms of the layers, which in practice is extremely loose. Virmaux & Scaman (2018) derive a tighter bound via a convex maximization problem; however the bound is typically intractable and can only be approximated. Combettes & Pesquet (2019) bound the Lipschitz constant of DNNs by noting that the common nonlinearities employed as activation functions are averaged, nonexpansive operators; however, their method scales exponentially with depth of the network. (Zou et al., 2019) propose linear-program-based bounds specific to convolutional networks, which in practice are several orders of magnitude larger than empirical lower bounds. Upper bounds based on semidefinite programs which relax the quadratic constraints imposed by the nonlinearities are studied by Fazlyab et al. (2019); Raghunathan et al. (2018); Jin & Lavaei (2018). The bounds can be tight in practice but expensive to compute for deep networks; as such, Fazlyab et al. (2019) propose a sequence of SDPs which trade off computational complexity and accuracy. This allows us to compare our monDEQ bounds to their SDP bounds for networks of increasing depth (see Section 5). Latorre et al. (2019) show that the complexity of the optimization problems can be reduced by taking advantage of the typical sparsity of connections common to DNNs, but the resulting methods are still prohibitively expensive for deep networks.

**DEQs and monotone DEQs** An emerging focus of deep learning research is on *implicit-depth* models, typified by Neural ODEs (Chen et al., 2018) and deep equilibrium models (DEQs) (Bai et al., 2019; 2020). Unlike traditional deep networks which compute their output by sequential, layer-wise computation, implicit-depth models simulate "infinite-depth" networks by specifying, and directly solving for, some analytical conditions satisfied by their output. The DEQ model directly solves for the fixed-point of an infinitely-deep, weight-tied and input-injected network, which would consist of the iteration $z_{i+1} = g(z_i, x)$ where, $g$ represents a nonlinear layer computation which is applied repeatedly, $z_i$ is the activation at "layer" $i$, and $x$ is the network input, which is injected at each layer. Instead of iteratively applying the function $g$ (which indeed may not converge), the infinite-depth fixed-point $z^* = g(z^*, x)$ can be solved using a root-finding method. A key advantage of DEQs is that backpropagation through the fixed-point can be performed analytically using the implicit function theorem, and DEQ training therefore requires much less memory than DNNs, which need to store the intermediate layer activations for backpropagation.

In standard DEQs, existence of a unique fixed point is not guaranteed, nor is stable convergence to a fixed-point easy to obtain in practice. Monotone DEQs (monDEQs) (Winston & Kolter, 2020) improve upon this aspect by parameterizing the DEQ in a manner that guarantees the existence of a stable fixed point. Monotone operator theory provides a class of *operator splitting methods* which are guaranteed to converge linearly to the fixed point (see Ryu & Boyd (2016) for a primer). The

monDEQ considers a weight-tied, input-injected network with iterations of the form

$$z^{(k+1)} = \sigma(W z^{(k)} + Ux + b) \tag{1}$$

where $x \in \mathbb{R}^n$ is the input, $U \in \mathbb{R}^{h \times n}$ the input-injection weights, $z^{(i)} \in \mathbb{R}^h$ the hidden unit activations at "layer" $i$, and $W \in \mathbb{R}^{h \times h}$ the hidden-unit weights, and $b \in R^h$ a bias term, and $\sigma : \mathbb{R}^h \to \mathbb{R}^h$ an elementwise nonlinearity. The output of the monDEQ is defined as the fixed point of the iteration, a $z^*$ such that

$$z^* = \sigma(W z^* + Ux + b). \tag{2}$$

Just as for DEQs, forward iteration of this system need not converge to $z^*$; instead, the fixed point is found as the solution to a particular operator splitting problem. Various operator splitting methods can be employed here, for example *forward-backward* iteration, which results in a damped version of the forward iteration

$$z^{(k+1)} = \sigma(z^{(k)} - \alpha((I - W)z^{(k)} - (Ux + b))) = \sigma((I - \alpha(I - W))z^{(k)} + \alpha(Ux + b)). \tag{3}$$

The operator $I - \alpha(I - W)$ appearing in this iteration is *contractive* for any $0 < \alpha \le 2m/L^2$, and this iteration is guaranteed to converge so long as the operator $I - W$ is Lipschitz and strongly monotone with parameters $L$ (which is in fact the spectral norm $\|I - W\|_2$) and $m$ (Ryu & Boyd, 2016). In Section 3, we will see how unrolling this iteration leads directly to a bound on the Lipschitz constant of the monDEQ. To ensure the strong monotonicity condition, that $I - W \succeq mI$, the monDEQ parameterizes $W$ as

$$W = (1 - m)I - A^T A + B - B^T.$$

The strong-monotonicity parameter $m$ will in fact figure in directly to the Lipschitz constant of the monDEQ.

**Lipschitz constants for implicit-depth models** A few prior works have proposed methods for bounding the Lipschitz constants of other classes of implicit depth network. Ghaoui et al. (2020) define restrictive conditions for well-posedness of an implicit network which are different from those of the monDEQ. In particular, they require the weight matrix $W$ to be such that forward iteration is stable (as opposed to the stability of the operator splitting methods required by monDEQ). They derive Lipschitz constants and robustness guarantees under these conditions; for example when $\|W\|_\infty < 1$, then a Lipschitz bound can be derived by simply manipulating the fixed-point equation as they demonstrate in equation 4.3. Herrera et al. (2020) propose a framework for implicit depth models which incorporate the Neural ODE (Chen et al., 2018) (which is the solution of an ODE at a given time T) but not the monDEQ (which can be cast as finding the *equilibrium point* of an ODE). They derive bounds on the Lipschitz constant w.r.t. network weights, but their framework cannot be applied to bound the Lipschitz constant of the monDEQ.

## 3 LIPSCHITZ CONSTANTS FOR MONOTONE DEQS

We now present our main methodological contributions, easily-computable bounds on the Lipschitz constants of monDEQs. We first derive the Lipschitz bound on the input-output mapping defined by the monDEQ, followed by that for the weight-output mapping. As we describe below, both bounds turn out to depend inversely on the strong-monotonicity parameter $m$ of the monDEQ. Since $m$ is chosen for the monDEQ at design time, this implies an analytical handle on its Lipschitz constant.

### 3.1 LIPSCHITZ CONSTANTS WITH RESPECT TO INPUT

The naive way of computing $L$ for feedforward deep networks is by multiplying the spectral norms of the weight matrices. As stated above, just employing forward iterations does not lead to convergence of the monDEQ. Analogously, if we were to adopt the naive method and simply unroll the forward iterations of the monDEQ as described in equation 1, we would end up with an infinite product of spectral norms, which would not converge unless $W$ itself is contractive. Here again, we consider unrolling the averaged operator $T := I - \alpha(I - W)$ employed in the forward-backward iterations, which ensures that the monDEQ converges, and will also lead to a finite bound on the Lipschitz constant. Notice that $T$ appears in the forward iterations in equation 3. In the sequel, let $L[A]$ denote the Lipschitz constant of a function or operator $A$. The following proposition, which we prove in Appendix A, bounds the Lipschitz constant $L[T]$.

**Proposition 1.** $L[T] \leq \sqrt{1 - 2\alpha m + \alpha^2 L[I - W]^2}$

This implies that for $\alpha \in \left(0, \frac{2m}{L[I-W]^2}\right)$, $L[T] < 1$. In our subsequent analysis, we only consider values of $\alpha$ in this range. We are now ready to state our bound for the Lipschitz constant of the monDEQ:

**Theorem 1** (Lipschitz constant of monDEQ). *Let $f(x) = z^*$ denote the output of the monDEQ on input $x$, as in equation 2. Consider any $x, y \in \mathbb{R}^n$. Then, we have that*

$$\|f(x) - f(y)\|_2 \leq \frac{\|U\|_2}{m} \|x - y\|_2.$$

*In other words, $L[f] \leq \frac{\|U\|_2}{m}$.*

*Proof.* Let $f_k(x) = z^{(k)}$ denote the $k^{th}$ iterate of the forward-backward iterations as described in equation 3 (we begin with $f_0(x) = 0$). We will try and unroll these iterations in the following:

$$
\begin{aligned}
\|f_k(x) - f_k(y)\|_2 &= \|\sigma(Tf_{k-1}(x) + \alpha Ux + \alpha b) - \sigma(Tf_{k-1}(y) + \alpha Uy + \alpha b)\|_2 \\
&\leq \|Tf_{k-1}(x) + \alpha Ux + \alpha b - Tf_{k-1}(y) - \alpha Uy - \alpha b)\|_2 \quad (\sigma = \text{ReLU is 1-Lipschitz}) \\
&= \|T(f_{k-1}(x) - f_{k-1}(y)) + \alpha U(x - y)\|_2 \leq \|T(f_{k-1}(x) - f_{k-1}(y))\|_2 + \alpha \|U(x-y)\|_2 \\
&\leq L[T]\|f_{k-1}(x) - f_{k-1}(y)\|_2 + \alpha L[U]\|x - y\|_2 \\
&\leq L[T]^k \|f_0(x) - f_0(y)\|_2 + \alpha \|U\|_2 \|x - y\|_2 \cdot \sum_{i=0}^{k-1} (L[T])^i \quad \text{(unrolling } k \text{ times)} \\
&= \alpha \|U\|_2 \|x - y\|_2 \cdot \sum_{i=0}^{k-1} (L[T])^i \quad \text{(since } f_0(x) = f_0(y) = 0)
\end{aligned}
$$

Since the above inequality holds for all $k$, we can take the limit on both sides as $k \to \infty$, keeping $\alpha$ fixed. But notice that since the forward-backward iterations converge to the true $f$ (which does not depend on $\alpha$), we have that $\lim_{k\to\infty} f_k = f$. That is, the dependence on $\alpha$ disappears on the LHS once we take the limit on $k$. Thus, by using the continuity of the $l_2$ norm, we have

$$
\begin{aligned}
\|f(x) - f(y)\|_2 &= \left\| \lim_{k\to\infty} f_k(x) - \lim_{k\to\infty} f_k(y) \right\|_2 \leq \alpha \|U\|_2 \|x - y\|_2 \cdot \sum_{i=0}^{\infty} (L[T])^i \\
&= \frac{\alpha \|U\|_2}{1 - L[T]} \|x - y\|_2 \quad \text{(since } L[T] < 1) \\
&\leq \frac{\alpha \|U\|_2}{1 - \sqrt{1 - 2\alpha m + \alpha^2 L[I - W]^2}} \|x - y\|_2 \quad \text{(from Proposition 1)}
\end{aligned}
$$

Now, since the above result holds for any $\alpha$ in the range considered, taking $\alpha \to 0$, we have that

$$
\begin{aligned}
L[f] &\leq \lim_{\alpha \to 0} \frac{\alpha \|U\|_2}{1 - \sqrt{1 - 2\alpha m + \alpha^2 L[I - W]^2}} \\
&= \frac{\|U\|_2}{m} \quad \text{(applying L'Hopital's rule)}
\end{aligned}
$$

$\square$

We observe here that the Lipschitz constant of the monDEQ with respect to its inputs indeed depends on only two quantities, namely $\|U\|_2$ and $m$, and doesn't depend at all on the weight matrix $W$. Furthermore, because $m$ is a hyperparameter chosen by the user, this illustrates that monDEQs have the notable property that one can essentially control the Lipschitz parameter of the network (insofar as the influence of $W$ is concerned) by appropriately choosing $m$, and not require any additional structure or regularization on $W$. This is in stark contrast to most existing DNN architectures, where enforcing Lipschitz bounds requires substantial additional effort.

## 3.2 LIPSCHITZ CONSTANTS WITH RESPECT TO WEIGHTS

We now turn to the question of bounding the change in the output of the monDEQ when the *weights* are perturbed but the input remains fixed. This calculation has several important use cases, one of which is in the derivation of generalization bounds for the monDEQ. Given a bound on the change in the output on perturbing the weights of the monDEQ, we can derive bounds on the generalization error in a straight-forward manner, as detailed in Section 4 below. The following theorem establishes a perturbation bound for the monDEQ.

**Theorem 2** (Perturbation bound for monDEQ). *Let $I - W \succeq mI$ and $I - \bar{W} \succeq \bar{m}I$. The change in the output of the monDEQ on perturbing the weights and biases from $W, U, b$ to $\bar{W}, \bar{U}, \bar{b}$ is bounded as follows:*

$$\|f(\bar{W}, \bar{U}, \bar{b}) - f(W, U, b)\|_2 \leq \frac{\|\bar{W} - W\|_2 \|Ux + b\|_2}{m\bar{m}} + \frac{\|(\bar{U} - U)x\|_2 + \|\bar{b} - b\|_2}{\bar{m}}$$

The proof steps for Theorem 2 parallel closely those involved in the derivation of the Lipschitz constant with respect to the inputs, and are outlined in Appendix B. We highlight here again that the bound depends inversely on $m$, a design parameter in our control. Further, when compared to a similar perturbation bound derived in Neyshabur et al. (2018), we note that our perturbation bound for the monDEQ *does not* involve a depth-dependent product of spectral norms of weights. In addition, although we state the theorem in terms of a perturbation of $W$ (which can thus lead to a different strong monotonicity parameter $\bar{m}$), the bound can also be adapted to perturbations on $A$ and $B$ in the typical monDEQ parameterization, which leads to a perturbed network that will necessarily still have the same monotonicity parameter $m$ as the original (indeed, we take this approach in the next section, when deriving the generalization bound).

## 4 GENERALIZATION BOUND FOR MONDEQ

In this section, we demonstrate how the perturbation bound derived in Section 3.2 leads directly to a deterministic PAC-Bayes, margin-based bound on the monDEQ generalization error, following the analysis for DNNs of Neyshabur et al. (2018). A key difference from our work, however, is that the perturbation bound they derive involves the product of spectral norms of all the weight matrices in the DNN. Thus, as the network gets deeper, their bound grows exponentially looser. As in Neyshabur et al. (2018), our generalization bound is based on two key ingredients. The first is their deterministic PAC-Bayes margin bound (Lemma 1 in the Appendix C), which adapts traditional PAC-Bayes bounds to bound the the expected risk of a parameterized, deterministic classifier in terms of its empirical margin loss. The second is the perturbation bound on monDEQ with respect to weights as derived in Section 3.2 above. Crucially, since our perturbation bound does not explicitly involve a product of spectral norms of weights (which in the case of the monDEQ, would be an infinite product), our final generalization bound does not either.

The monDEQ model we consider here consists of a fully connected layer at the end that maps $f$ to the output, so that $f_o(x) = W_o f(x) + b_o$, where $W_o$ and $b_o$ are the weights and bias in the output layer; these parameters are important to include here since they contribute directly to the perturbation bound. We also restrict the input $x$ to the monDEQ to lie in an $l_2$ norm ball of radius $B$. Let $h$ denote the hidden dimension of the monDEQ, and $M$ the size of the training set, and define $\beta := \max\{\|U\|_2, \|A\|_2, \|b\|_2, \|W_o\|_2\}$. Let $L_\gamma(f_o)$ denote the expected margin loss at margin $\gamma$ of the monDEQ on the data distribution $\mathcal{D}$, where

$$L_\gamma(f_o) = \mathbb{P}_{(x,y) \sim \mathcal{D}} \left[ f_o(x)_y \leq \gamma + \max_{j \neq y} f_o(x)_j \right]$$

and $\hat{L}_\gamma(f_o)$ denote the corresponding empirical margin loss on the training dataset. We are now ready to state our generalization bound for the monDEQ:

**Theorem 3** (Generalization bound for monDEQ). *Let*

$$\sum \|W_\cdot\|_F^2 = \|A\|_F^2 + \|B\|_F^2 + \|U\|_F^2 + \|b\|_F^2 + \|W_o\|_F^2 + \|b_o\|_F^2$$

*For any $\delta, \gamma > 0$, with probability at least $1 - \delta$ over the training set of size $M$, we have that*

$$L_0(f_o) \leq \hat{L}_\gamma(f_o) + \mathcal{O}\left( \sqrt{\frac{h \ln(h)[\beta^2 B(\gamma + \beta) + m\beta B + m^2]^2}{\gamma^2 m^4 M} \sum \|W_\cdot\|_F^2 + \frac{\ln(\frac{M\sqrt{M}}{\delta})}{M}} \right)$$

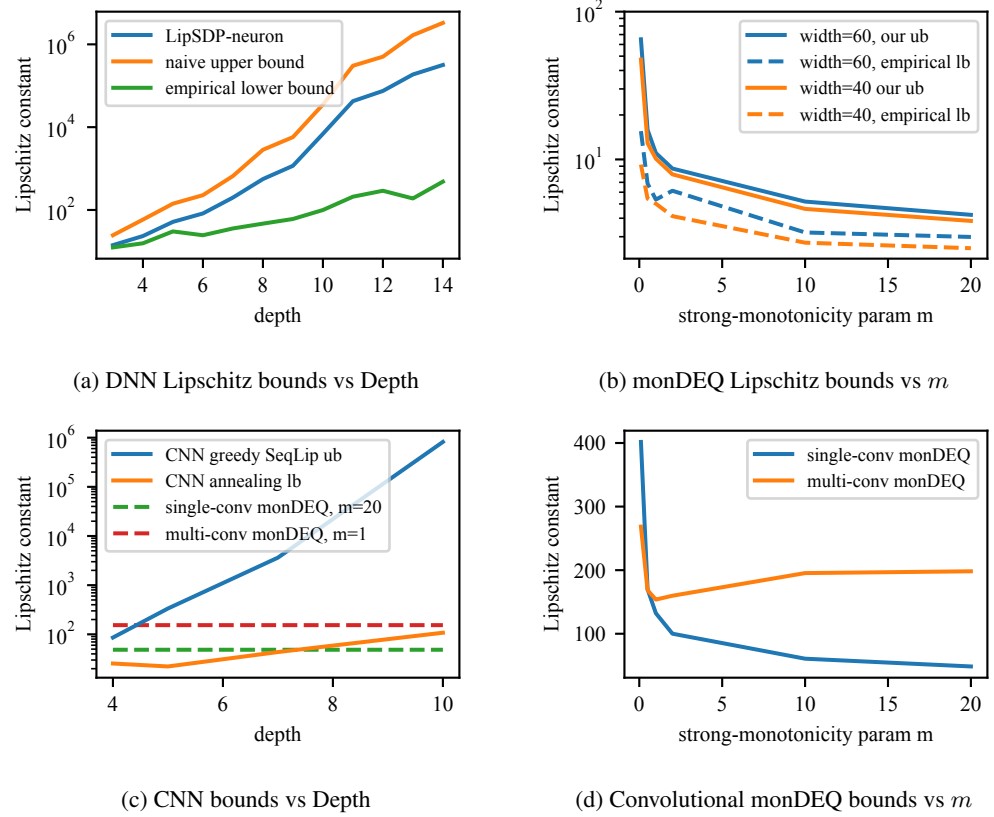

(a) DNN Lipschitz bounds vs Depth

(b) monDEQ Lipschitz bounds vs $m$

(c) CNN bounds vs Depth

(d) Convolutional monDEQ bounds vs $m$

Figure 1: MNIST results: Lipschitz bounds as a function of depth and strong monotonicity parameter. lb: lower bound; ub: upper bound.

Note that our bound above does not involve any depth-like term that scales exponentially, like the term that involves the product of spectral norms of the weight matrices in Neyshabur et al. (2018) (while still having the same dependence on $h$, which is $\sqrt{h \ln h}$). To the best of our knowledge, this is the first generalization bound for an implicit-layer model having effectively infinite depth. The proof of Theorem 3 is given in Appendix C.

# 5 EXPERIMENTAL RESULTS[1]

## 5.1 LIPSCHITZ CONSTANTS

In this section, we empirically verify the tightness of the Lipschitz constant of the monDEQ with respect to inputs. We conduct all our experiments on MNIST and CIFAR-10, for which several benchmarks exist for computing the Lipschitz constant. We conduct experiments for different monDEQ architectures (fully connected/convolutional) with varying parameters (strong-monotonicity parameter $m$ and width $h$), which we compare to DNNs with different depths and widths. We compute empirical lower bounds by maximizing the norm of the gradient at 10k randomly sampled points. A naive upper bound can be computed as $\prod_{i=1}^{d} \|W_i\|_2$. We include these bounds wherever applicable.

**MNIST** Here, we train DNNs for various depths from $d = 3, 4, \ldots, 14$ for a fixed hidden layer width $h = 40$, and plot (Figure 1a) the bound on the Lipschitz constant given by the SDP-based method of Fazlyab et al. (2019) on these DNNs. We can observe that all estimates of the Lipschitz constant increase exponentially with depth. For comparison, in Figure 1b we plot our Lipschitz constant bounds for monDEQs with fixed $h = 40, 60$, for a range of strong-monotonicity parameters $m$. We note that the DNNs all have test error of around 3%, while the monDEQ test error ranges from

---

[1]Experimental code available at `https://github.com/locuslab/lipschitz_mondeq`.

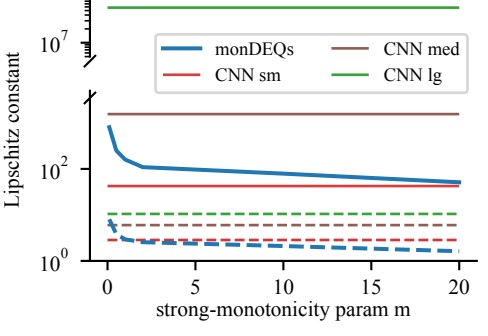

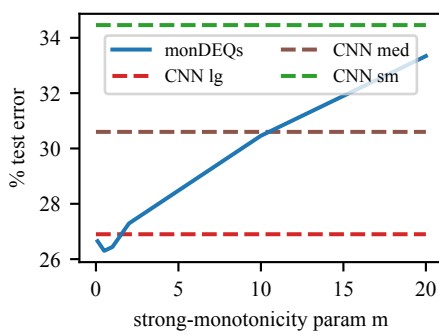

(a) Lipschitz bounds for monDEQs vs $m$ and for three CNN models. Solid lines: upper bounds; dashed lines: empirical lower bounds.

(b) Test error for CNNs and monDEQs vs $m$.

Figure 2: CIFAR-10 results: Lipschitz bounds and test accuracy for monDEQs as a function of strong monotonicity parameter and for CNNs. See text for description of models.

2.4%-4.3%, increasing with $m$ (see Figure 5 in Appendix F for details). We see that the Lipschitz constant of the monDEQ is much smaller, and that on increasing $m$, the Lipschitz constant of the monDEQ decreases, outlining how we can exercise control on the Lipschitz constant.

We also compare the Lipschitz constants of monDEQs and DNNs having the same width, for a fixed depth $d = 5$. The results are shown in Figure 6 in Appendix F. The DNN numbers are derived from Figure 2(a) in (Fazlyab et al., 2019). We observe that the Lipschitz constant of the monDEQ for the same width (and essentially infinite depth) is much lower than the bounds for regular DNNs.

Next, using the bound derived in Section 3.1, we compute the Lipschitz constant of convolutional monDEQ architectures, namely single convolutional and multi-tier convolutional monDEQs. We compare to the numbers in Figure 5 in Virmaux & Scaman (2018), which reports the Lipschitz constants computed by various methods for different CNNs with increasing depth. For our estimate on the single convolutional monDEQ, we use a single convolutional layer with 128 channels, whereas for the multi-tier convolutional monDEQ, we use 3 convolutional layers with 32, 64 and 128 channels. In Figure 1c, we can observe that as for DNNs, the CNN Lipschitz constants estimated by existing methods also suffer with depth. However, we can observe in Figure 1d that the Lipschitz bounds for convolutional monDEQs are much smaller. Also, on increasing $m$, we can control the Lipschitz constant of both single as well as multi-tier convolutional monDEQs. The test error for the convolutional monDEQs is 0.65%-3.22%, increasing with $m$ (see Figure 5 in Appendix F), but is not reported for the CNNs in Virmaux & Scaman (2018).

**CIFAR-10** To demonstrate that the Lipschitz bounds scale to larger datasets, we run similar experiments on CIFAR-10. Figure 2a shows our bound (solid blue lines) for single-convolutional monDEQs (128 channels) with a range of $m$ values, together with empirical lower bounds (dashed blue lines). Also shown are upper bounds (solid lines) and empirical lower bounds (dashed lines) for three standard CNN models (CNN sm, med, and lg, having 2, 4, and 6 convolutions respectively, detailed in Appendix F). The upper bounds are computed using the Greedy SeqLip method of Virmaux & Scaman (2018). We see that a) the monDEQ Lipschitz bounds decrease with $m$, and b) the upper bounds (and gap between upper and lower bounds) for the medium and large CNNs are large by comparison. In Figure 2b we see that test error of the monDEQ increases with $m$ from 26% to 33%, and that, despite their much higher Lipschitz bounds, the three CNNs have similar test error.

Finally, on CIFAR-10 with data augmentation, we also trained a larger multi-tier convolutional monDEQ with three convolutional layers with 64, 128, and 128 channels. This model obtains 10.25% test error and has a Lipschitz upper bound of 1996.86, which is on par with the upper bound of the medium CNN (plotted in brown; with $L = 1554.01$, test error = 30.6%).

**Unrolling monDEQs** In this experiment, we study if unrolling the monDEQ with $m = 1, h = 40$ up to a finite depth and constructing an equivalent DNN with this depth leads to a tight estimate of

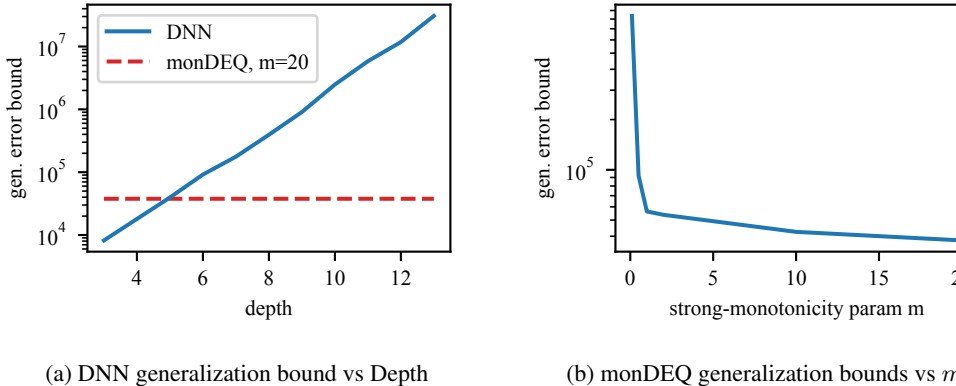

(a) DNN generalization bound vs Depth        (b) monDEQ generalization bounds vs $m$

Figure 3: Generalization bounds for DNNs and monDEQs as a function of depth and $m$.

the Lipschitz constant of the monDEQ. Concretely, we do this for two operator splitting methods in the monDEQ: Forward-backward (FB) iterations and Peaceman-Rachford (PR) iterations. For each value of $\alpha$ in a range, we calculate the number of iterations (FB or PR) required to converge within a tolerance 1e-3, and construct the equivalent DNN with this depth. Note that these unrolled DNNs compute the same function as the monDEQ (up to tolerance), and therefore, must have the same Lipschitz constant (which is around 10). We compute naive upper bounds on the Lipschitz constants of these DNNs (we cannot use the SDP-based bound of Fazlyab et al. (2019) due to technicalities in the construction of the unrolled DNN; refer to Appendix D). We can observe (Figure 7 in Appendix F) that the upper bounds corresponding to both PR and FB iterations are in the range $10^5$ to $10^{13}$, suggesting that unrolling the monDEQ and employing standard techniques on the unrolled monDEQ is not a viable way to bound the Lipschitz constant. More details about the construction of these equivalent DNNs for both FB and PR iterations are provided in Appendix D.

## 5.2 GENERALIZATION BOUNDS

A key advantage of the monDEQ generalization bounds derived in Section 4 is the lack of any depth analog that can cause the bounds to grow exponentially. To assess this aspect experimentally, we first compute the DNN generalization bound following the protocol of Nagarajan & Kolter (2018). We train DNNs (width = 40) of varying depth of 3 to 14 layers, and compare to similar monDEQs with various $m$ values. Each model is trained on a sample of 4096 MNIST examples until the margin error at margin $\gamma = 10$ reaches below $10\%$ which serves to standardize the experiments across choice of batch size and learning rate. As widely reported, we see that DNN bounds increase exponentially with depth, ranging numerically from $10^4$ for depth 3 networks to $10^8$ (see Figure 3a). For monDEQs of width = 40, the bound decreases monotonically with $m$, and is confined to the range $10^4$ to $10^6$, as seen in Figure 3b (note the difference in scale). In contrast, the true test error of the DNNs increases only slightly with depth, and that of the monDEQs increases only slightly with $m$. Note that the DNNs and monDEQ s have comparable test error (see Figure 8 in Appendix G).

Finally, as done for Lipschitz bounds above, we compare our generalization bound to what we obtain by unrolling the monDEQ into a DNN, and then computing the Neyshabur et al. (2018) bound more-or-less directly (see Appendix E). We do this only for FB iterations, as the inverted operators of PR iterations complicate the analysis. As seen in Figure 8c, the resulting bounds are quite high, though the difference with our bound is not as great as was seen for the unrolled Lipschitz bounds above. We attribute this to the fact that our generalization bound technique is a minimal modification to that of Neyshabur et al. (2018); we expect that it can be tightened with more refined analysis.

## 5.3 ADVERSARIAL ROBUSTNESS OF MONDEQS

In this section, we empirically demonstrate an important use case for the tight Lipschitz constants of the monDEQ: robustness to adversarial examples. We experiment with both certified adversarial $\ell_2$ robustness as well as empirical robustness to adversarial PGD $\ell_2$-bounded attacks. Here we describe our results on MNIST, and report similar experiments on CIFAR-10 in Appendix H.

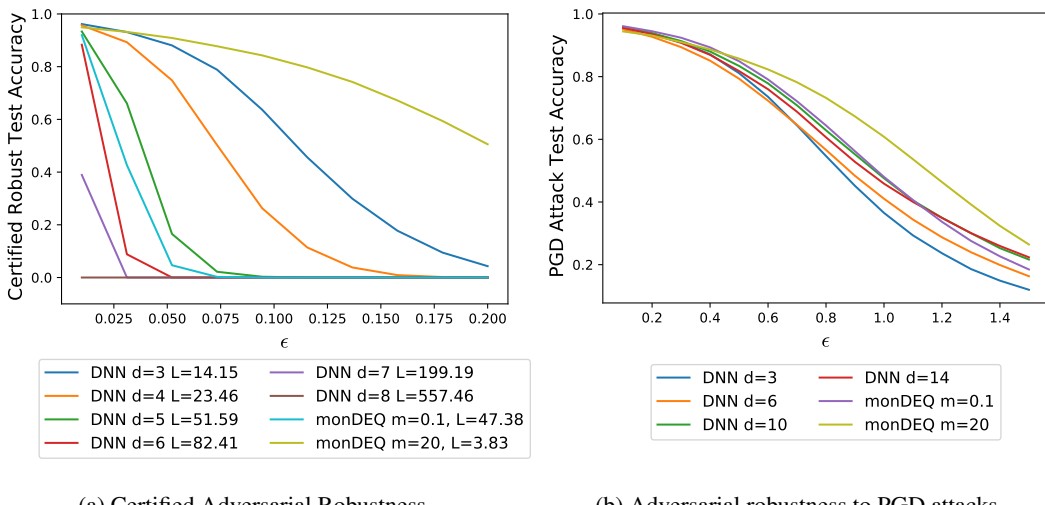

(a) Certified Adversarial Robustness    (b) Adversarial robustness to PGD attacks

Figure 4: Superior adversarial robustness of monDEQs as compared to DNNs

**Certified adversarial robustness**    Consider any point $x'$ within an $l_2$ ball of radius $\epsilon$ around $x$. Then, we have that

$$\|f(x) - f(x')\|_\infty \le \|f(x) - f(x')\|_2 \le L\|x - x'\|_2 \le L\epsilon$$

Define margin$(x) = f(x)_y - \max_{i \neq y} f(x)_i$, where $f(x)_y$ is the logit corresponding to the label $y$ of the input $x$. Then if $L\epsilon \le \frac{1}{2}$margin$(x)$, we are certified robust to any perturbed input $x'$ within an $l_2$-ball of radius $\epsilon$ around $x$. Thus, we can empirically compare DNNs and monDEQs with regards to this certificate on MNIST. For a range of $\epsilon$ values, we compute the (certified) robust test accuracy (fraction of points in test set for which the aforementioned condition holds) for trained DNNs as well as monDEQs. We note here that our choice of $\epsilon$ values corresponds to inputs normalized in the range $[0, 1]$. For DNNs, just as in Figure 1a, we vary the depth for fixed $h = 40$, and use the $L$ values computed by using the method in Fazlyab et al. (2019). For monDEQs, we set width $h = 40, m = 0.1, 20$ and substitute our upper bound for $L$. Note that since the $L$ values for monDEQs observed in Section 5.1 were significantly smaller than those for the DNNs, one would expect the condition for the certificate to hold more easily for monDEQs. Indeed, we verify this in our experiments. In Figure 4a, we can observe that the robust test accuracy for the monDEQ with $m = 20$ at $\epsilon = 0.2$ is 51%, while that for the best DNN $(d = 3)$, is just 4%. This illustrates that monDEQs allow for better certificates to adversarial robustness, owing to their small Lipschitz constants, and the ability to control it by setting $m$.

**Empirical robustness**    We also assess the empirical robustness of monDEQs to $l_2$-bounded Projected Gradient Descent attacks (implemented as part of the Foolbox toolbox (Rauber et al., 2017; 2020)) on both, trained monDEQs and DNNs on MNIST, and compute the accuracy on these adversarially perturbed test examples. Figure 4b shows the results: in general, over a range of $\epsilon$ values, the robust test accuracy of the monDEQ with $m = 20$ is larger than that of the DNNs.

## 6    CONCLUSION

In this paper, we derived Lipschitz bounds for monotone DEQs, a recently proposed class of impicit-layer networks, and showed that they depend in a straighforward manner on the strong monotonicity parameter $m$ of these networks. Having derived a Lipschitz bound with respect to perturbation in the weights, we were able to derive a PAC-Bayesian generalization bound for the monotone DEQ, which does not depend exponentially on depth. We showed empirically that our bounds are sensible, can be controlled by choosing $m$ suitably, and do not suffer with increasing depth of the network. As future work, we aim to analyze the vacuousness of the derived generalization bound. As such, since our bound does not suffer exponentially with depth, we hope to be able to make the analysis tighter and derive a non-vacuous generalization bound.

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

## A   PROOF OF PROPOSITION 1

*Proof.*

$$
\begin{aligned}
\|Tx - Ty\|_2^2 &= \|((1-\alpha)I + \alpha W)x - ((1-\alpha)I + \alpha W)y\|_2^2 \\
&= \|x - y - \alpha(I - W)(x - y)\|_2^2 \\
&= \|x - y\|_2^2 - 2\alpha(x - y)^T(I - W)(x - y) + \alpha^2\|(I - W)(x - y)\|_2^2 \\
&\leq \|x - y\|_2^2 - 2\alpha(x - y)^T(I - W)(x - y) + \alpha^2 L[I - W]^2\|x - y\|_2^2
\end{aligned}
$$

Now, note that by the strong monotonicity of the monDEQ,

$$
I - W \succeq mI,
$$

which implies that $(x - y)^T(I - W)(x - y) \geq m\|x - y\|_2^2$. Substituting this bound above, we have that

$$
\begin{aligned}
\|Tx - Ty\|_2^2 &\leq \|x - y\|_2^2 - 2\alpha m\|x - y\|_2^2 + \alpha^2 L[I - W]^2\|x - y\|_2^2 \\
&= (1 - 2\alpha m + \alpha^2 L[I - W]^2)\|x - y\|_2^2
\end{aligned}
$$

Thus, we have that $L[T] \leq \sqrt{1 - 2\alpha m + \alpha^2 L[I - W]^2}$                                     □

## B   PROOF OF THEOREM 2

In order to derive the perturbation bound in Theorem 2, we first state the following proposition, which bounds the norm of the output after $k$ forward-backward iterations.

**Proposition 2.** *Let $f_k(W, U, b)$ denote the $k^{th}$ iterate of the forward-backward iterations of the monDEQ parameterized by $W, U, b$ on a fixed arbitrary input $x$. Further, let $T(W) = (1 - \alpha)I + \alpha W$. Then, we have that*

$$
\|f_k(W, U, b)\|_2 \leq \frac{\alpha\|Ux + b\|_2}{1 - L[T(W)]}
$$

*Proof of Proposition 2.*

$$
\begin{aligned}
\|f_k(W, U, b)\|_2 &= \|\sigma(Tf_{k-1}(W, U, b) + \alpha(Ux + b))\|_2 \\
&\leq \|Tf_{k-1}(W, U, b) + \alpha(Ux + b)\|_2 \quad \text{(ReLU is 1-Lipschitz)} \\
&\leq \|T\|_2\|f_{k-1}(W, U, b)\|_2 + \alpha\|Ux + b\|_2 \\
&= L[T]\|f_{k-1}(W, U, b)\|_2 + \alpha\|Ux + b\|_2 \\
&\leq L[T]^k\|f_0(W, U, b)\|_2 + \alpha\|Ux + b\|_2 \sum_{i=0}^{k-1} L[T]^i \\
&= \alpha\|Ux + b\|_2 \sum_{i=0}^{k-1} L[T]^i \quad \text{(since } f_0(W, U, b) = 0) \\
&\leq \alpha\|Ux + b\|_2 \sum_{i=0}^{\infty} L[T]^i \\
&= \frac{\alpha\|Ux + b\|_2}{1 - L[T]} \quad \text{(since } L[T] < 1)
\end{aligned}
$$

□

We are now ready to prove Theorem 2.

*Proof of Theorem 2.* Denote $\bar{f}_k = f_k(\bar{W}, \bar{U}, \bar{b})$ and $f_k = f_k(W, U, b)$. Further, denote $\Delta_k = \|\bar{f}_k - f_k\|_2$. For $\alpha \in \left(0, \min\left(\frac{2m}{L[I - W]^2}, \frac{2\bar{m}}{L[I - \bar{W}]^2}\right)\right)$, we have from Proposition 1 that both

$\|T(W)\|_2, \|T(\bar{W})\|_2 < 1$. Thus,

$$\Delta_k = \|\sigma[T(\bar{W})\bar{f}_{k-1} + \alpha(\bar{U}x + \bar{b})] - \sigma[T(W)f_{k-1} + \alpha(Ux + b)]\|_2$$
$$\leq \|T(\bar{W})\bar{f}_{k-1} - T(W)f_{k-1} + \alpha(\bar{U} - U)x + \alpha(\bar{b} - b)\|_2$$
$$\leq \|T(\bar{W})(\bar{f}_{k-1} - f_{k-1}) + (T(\bar{W}) - T(W))f_{k-1} + \alpha(\bar{U} - U)x + \alpha(\bar{b} - b)\|_2$$
$$\leq \|T(\bar{W})(\bar{f}_{k-1} - f_{k-1}) + \alpha(\bar{W} - W)f_{k-1} + \alpha(\bar{U} - U)x + \alpha(\bar{b} - b)\|_2$$
$$\leq \|T(\bar{W})\|_2 \Delta_{k-1} + \alpha\|\bar{W} - W)\|_2\|f_{k-1}\|_2 + \alpha\|(\bar{U} - U)x\|_2 + \alpha\|\bar{b} - b\|_2$$
$$\leq \|T(\bar{W})\|_2 \Delta_{k-1} + \frac{\alpha^2\|\bar{W} - W)\|_2\|Ux + b\|_2}{1 - \|T(W)\|_2} + \alpha\|(\bar{U} - U)x\|_2 + \alpha\|\bar{b} - b\|_2 \quad \text{(Proposition 2)}$$
$$\leq \left(\frac{\alpha^2\|\bar{W} - W)\|_2\|Ux + b\|_2}{1 - \|T(W)\|_2} + \alpha\|(\bar{U} - U)x\|_2 + \alpha\|\bar{b} - b\|_2\right)\sum_{i=0}^{k-1} \|T(\bar{W})\|_2^i$$

Notice here again that the above inequality holds for all $k$. Taking the limit as $k \to \infty$ similar to the step in the proof of Theorem 1, we have

$$\|f(\bar{W}, \bar{U}, \bar{b}) - f(W, U, b)\|_2 = \lim_{k \to \infty} \Delta_k$$
$$\leq \left(\frac{\alpha^2\|\bar{W} - W)\|_2\|Ux + b\|_2}{1 - \|T(W)\|_2} + \alpha\|(\bar{U} - U)x\|_2 + \alpha\|\bar{b} - b\|_2\right)\sum_{i=0}^{\infty} \|T(\bar{W})\|_2^i$$
$$= \frac{\alpha^2\|\bar{W} - W)\|_2\|Ux + b\|_2}{(1 - \|T(W)\|_2)(1 - \|T(\bar{W})\|_2)} + \alpha\frac{\|(\bar{U} - U)x\|_2 + \|\bar{b} - b\|_2}{1 - \|T(\bar{W})\|_2}$$

Finally, taking $\alpha \to 0$, we have that

$$\|f(\bar{W}, \bar{U}, \bar{b}) - f(W, U, b)\|_2 \leq \lim_{\alpha \to 0} \left(\frac{\alpha^2\|\bar{W} - W\|_2\|Ux + b\|_2}{(1 - \|T(W)\|_2)(1 - \|T(\bar{W})\|_2)} + \right.$$
$$\left. \alpha\frac{\|(\bar{U} - U)x\|_2 + \|\bar{b} - b\|_2}{1 - \|T(\bar{W})\|_2}\right)$$
$$= \|\bar{W} - W)\|_2\|Ux + b\|_2 \lim_{\alpha \to 0} \frac{\alpha}{1 - \|T(W)\|_2} \cdot \lim_{\alpha \to 0} \frac{\alpha}{1 - \|T(\bar{W})\|_2}$$
$$+ (\|(\bar{U} - U)x\|_2 + \|\bar{b} - b\|_2) \lim_{\alpha \to 0} \frac{\alpha}{1 - \|T(\bar{W})\|_2}$$
$$= \frac{\|\bar{W} - W\|_2\|Ux + b\|_2}{m\bar{m}} + \frac{\|(\bar{U} - U)x\|_2 + \|\bar{b} - b\|_2}{\bar{m}} \quad \text{(applying L'Hopital's rule)}$$

$\square$

## C   PROOF OF THEOREM 3

*Proof.* We first state Lemma 1 from Neyshabur et al. (2018).

**Lemma 1** (Lemma 1 from Neyshabur et al. (2018)). *Let $f_w$ be any predictor with parameters $w$, and let $P$ denote any distribution on the parameters that is independent of the training data. Then, for any $\delta, \gamma > 0$, with probability $\geq 1 - \delta$ over the training data of size $M$, for any $w$, and any random perturbation $u$ such that $\mathbb{P}[\max_x \|f_{w+u}(x) - f_w(x)\|_\infty < \frac{\gamma}{4}] \geq \frac{1}{2}$, we have*

$$L_0(f_w) \leq \hat{L}_\gamma(f_w) + 4\sqrt{\frac{KL(w + u||P) + \ln \frac{6M}{\delta}}{M - 1}}$$

Now, we derive a perturbation bound for the monDEQ when we incorporate the a fully connected layer at the end, as mentioned in Section 4. That is, we consider $f_o(x) = W_o f(x) + b_0$ where $f$ is the output of the fixed-point iterations of the monDEQ. Next, we consider perturbations $\Delta_A, \Delta_B, \Delta_U, \Delta_b, \Delta_{W_o}, \Delta_{b_o}$ for $A, B, U, b, W_o, b_o$ respectively. The entries in the perturbation matrices are each drawn independently from a Gaussian $\mathcal{N}(0, \sigma^2)$. Let $\bar{f}_o$ to denote the function at the

perturbed values of the weights. Then, we have that

$$
\begin{aligned}
\|\bar{f}_o(x) - f_o(x)\|_2 &= \|\bar{W}_o \bar{f}(x) + \bar{b}_o - W_o f(x) - b_o\|_2 \\
&\leq \|W_o(\bar{f}(x) - f(x)) + (\bar{W}_o - W_o)\bar{f}(x)\|_2 + \|\bar{b}_o - b_o\|_2 \\
&\leq \|W_o\|_2 \Delta + \frac{\|\bar{W}_o - W_o\|_2 \|Ux + b\|_2}{\bar{m}} + \|\bar{b}_o - b_o\|_2
\end{aligned}
$$

where $\Delta$ is the bound from Theorem 2.

Now, let $\beta = \max(\|U\|_2, \|A\|_2, \|W_o\|_2, \|b\|_2)$. Just as in (Neyshabur et al., 2018), since we cannot use $\beta$ in determining the parameters of the prior distribution $P$ in Lemma 1, we will consider predetermined values $\tilde{\beta}$ on a grid, and then do a union bound. For now, we fix $\tilde{\beta}$, and consider all the values $\beta$ such that $|\beta - \tilde{\beta}| < c_1 \beta$ for some constant $c_1 < 1$.

Since the entries in the perturbations are drawn from $\mathcal{N}(0, \sigma^2)$, we have the following bound on the $l_2$ norms of these perturbations:

$$
\mathbb{P}_{\Delta. \sim \mathcal{N}(0, \sigma^2 I)}[\|\Delta.\|_2 > t] \leq 2h e^{-t^2/2h\sigma^2}
$$

where $\Delta.$ is a placeholder for each of the perturbation matrices. Thus, we have that with probability $\geq 1/2$, all of the $\|\Delta.\|_2$ are bounded above by $\sigma\sqrt{2h\ln(24h)} := \omega$.

Now, we bound the perturbation in $\|W\|_2$ when $A$ and $B$ are perturbed. We have that

$$
\begin{aligned}
\|\Delta_W\|_2 &= \|A^T \Delta_A + \Delta_A^T A + \Delta_A^T \Delta_A + \Delta_B - \Delta_B^T\|_2 \\
&\leq 2\|A\|_2 \|\Delta_A\|_2 + \|\Delta_A\|_2^2 \\
&\leq 2\omega(\beta + \omega) \quad \text{(with probability } 1/2)
\end{aligned}
$$

Substituting this above, we have that for all $x$, with probability at least $1/2$,

$$
\|\bar{f}_o(x) - f_o(x)\|_2 \leq \frac{2\beta^2 \omega(\beta + \omega)(B + 1)}{m^2} + \frac{2\omega\beta(B+1)}{m} + \omega
$$

Here, let $c_2 > 0$ be some constant such that $\omega \leq c_2 \beta$. Thus, we have that

$$
\begin{aligned}
\|\bar{f}_o(x) - f_o(x)\|_2 &\leq \omega \left( \frac{2\beta^3(1 + c_2)(B + 1)}{m^2} + \frac{2\beta(B+1)}{m} + 1 \right) \\
&\leq \omega \left( \frac{2\tilde{\beta}^3(1 + c_2)(B + 1)}{m^2(1 - c_1)^3} + \frac{2\tilde{\beta}(B+1)}{m(1 - c_1)} + 1 \right) \\
&= \sigma\sqrt{2h\ln(24h)} \left( \frac{2\tilde{\beta}^3(1 + c_2)(B + 1) + 2m\tilde{\beta}(B+1)(1-c_1)^2 + m^2(1-c_1)^3}{m^2(1 - c_1)^3} \right)
\end{aligned}
$$

Setting $\sigma = \frac{\gamma m^2(1-c_1)^3}{4\sqrt{2h\ln(24h)}(2\tilde{\beta}^3(1+c_2)(B+1)+2m\tilde{\beta}(B+1)(1-c_1)^2+m^2(1-c_1)^3)}$ makes this $\leq \frac{\gamma}{4}$. Since we need $\omega \leq c_2\beta$ we can take the smallest $c_2$ such that

$$
c_2 \geq \frac{(1 + c_1)\sigma\sqrt{2h\ln(24h)}}{\tilde{\beta}} \geq \frac{\sigma\sqrt{2h\ln(24h)}}{\beta} = \frac{\omega}{\beta}
$$

Taking $c_2 = \frac{(1+c_1)\gamma}{4\tilde{\beta}}$ suffices. We will later plug in the value

$$
c_3 := \frac{(1 + c_1)\gamma}{4(1 - c_1)\beta} \geq c_2
$$

to bound $c_2$.

Then,

$$
\begin{aligned}
KL(W. + \Delta_{W.}|P) &\leq \frac{\sum \|W.\|_F^2}{2\sigma^2} \\
&= \frac{16h\ln(24h)(2\tilde{\beta}^3(1+c_2)(B+1) + 2m\tilde{\beta}(B+1)(1-c_1)^2 + m^2(1-c_1)^3)^2}{\gamma^2 m^4(1-c_1)^6} \sum \|W.\|_F^2 \\
&\leq \frac{16h\ln(24h)(2\beta^3(1+c_1)^3(1+c_3)(B+1) + 2m\beta(1+c_1)(B+1)(1-c_1)^2 + m^2(1-c_1)^3)^2}{\gamma^2 m^4(1-c_1)^6} \sum \|W.\|_F^2
\end{aligned}
$$

Then, by Lemma 1, we have that with probability $1 - \delta$,

$$L_0(f_0) \leq \hat{L}_\gamma(f_o)+$$

$$4\sqrt{\frac{16h\ln(24h)(2\beta^3(1+c_1)^3(1+c_3)(B+1) + 2m\beta(1+c_1)(B+1)(1-c_1)^2 + m^2(1-c_1)^3)^2}{\gamma^2 m^4 (1-c_1)^6 (M-1)} \sum \|W.\|_F^2 + \frac{\ln(\frac{6M}{\delta})}{M-1}}$$

Now we need to take a union bound over $\tilde{\beta}$ so that the above result holds for all $\beta$. Observe that we only need to consider $\beta$ in the range

$$\frac{\gamma m}{2(B+1)} \leq \beta \leq \frac{\gamma m \sqrt{M}}{2(B+1)}.$$

If $\beta \leq \frac{\gamma m}{2(B+1)}$ then $|f(x)| \leq \frac{\beta(B+1)}{m} \leq \frac{\gamma}{2}$ so $\hat{L}_\gamma(f) > 1$. If $\beta \geq \frac{\gamma m \sqrt{M}}{2(B+1)}$ then the second term in the first term in the numerator is greater than 1 so the theorem holds trivially.

So $|\beta - \tilde{\beta}| \leq c_1 \frac{\gamma m}{2(B+1)}$ guarantees $|\beta - \tilde{\beta}| \leq c_1\beta$ in this range. So we can use a cover of size $\sqrt{M}/2c_1$: This amounts to replacing $\ln 6M/\delta$ with $\ln 3M^{3/2}/c_1\delta$.

Finally, substituting $c_3 = \frac{(1+c_1)\gamma}{4(1-c_1)\beta}$, we get the theorem statement. We can also effectively optimize over $c_1$ to remove it from the final bound. $\square$

## D UNROLLING FORWARD-BACKWARD AND PEACEMAN-RACHFORD ITERATIONS

In this section, we derive the form of the equivalent feedforward DNN that computes the same quantity as running $k$ iterations of either Forward-Backward or Peaceman-Rachford iterations, which are operator-splitting methods for computing the fixed point of equation 1 (refer Winston & Kolter (2020)).

### D.1 UNROLLING FORWARD-BACKWARD ITERATIONS

One step in the Forward-Backward iterations computes

$$z^{(i+1)} = \sigma((1-\alpha)I + \alpha W)z^{(i)} + \alpha(Ux + b))$$

To simulate $k$ steps of these computations as a depth $k$-feedforward network, let us construct the following weight matrices:

$$W_1 = \begin{bmatrix} \alpha U \\ I \end{bmatrix}$$

$$W_u = \begin{bmatrix} (1-\alpha)I + \alpha W & \alpha U \\ 0 & I \end{bmatrix}$$

$$W_f = [W_o \quad 0]$$

Then, we can observe that

$$z^{(1)} = \sigma\left(W_1 x + \begin{bmatrix} \alpha b \\ 0 \end{bmatrix}\right)$$

$$z^{(i+1)} = \sigma\left(W_u \begin{bmatrix} z^{(i)} \\ x \end{bmatrix} + \begin{bmatrix} \alpha b \\ 0 \end{bmatrix}\right)$$

Here, $\sigma$ applies only to the top $h$ coordinates corresponding to $z^{(i)}$. Finally, we multiply the output of the monDEQ after $k$ iterations with the output weights, to obtain $f_o(x)$ as

$$f_o(x) = W_f \begin{bmatrix} z^{(k)} \\ x \end{bmatrix} + b_o$$

Thus, the equivalent depth-$k$ DNN which we construct would have weight matrices $W_1, [W_u]_{i=1}^{k-1}, W_f$. In this DNN, the ReLUs in the intermediate layers would only apply to the top coordinates corresponding to $z$ (this is the technical reason why we can't use the SDP bound given by (Fazlyab et al., 2019) for computing the Lipschitz constants of these unrolled networks, since they require the same nonlinearity to apply pointwise at all coordinates).

## D.2 Unrolling Peaceman-Rachford Iterations

Define $V = (I + \alpha(I - W))^{-1}$. One step in the Peaceman-Rachford iterations computes

$$u^{(i+1)} = u^{(i)} - 2z^{(i)} + 4Vz^{(i)} - 2Vu^{(i)} + 2\alpha VUx + 2\alpha Vb$$
$$z^{(i+1)} = \sigma(u^{i+1})$$

To simulate $k$ steps of these computations as a depth $k$-feedforward network, let us construct the following weight matrices:

$$W_1 = \begin{bmatrix} 2\alpha VU \\ 2\alpha VU \\ I \end{bmatrix}$$

$$W_u = \begin{bmatrix} I - 2V & 4V - 2I & 2\alpha VU \\ I - 2V & 4V - 2I & 2\alpha VU \\ 0 & 0 & I \end{bmatrix}$$

$$W_f = \begin{bmatrix} 0 & W_o & 0 \end{bmatrix}$$

Then, we can observe that

$$z^{(1)} = \sigma\left( W_1 x + \begin{bmatrix} 2\alpha Vb \\ 2\alpha Vb \\ 0 \end{bmatrix} \right)$$

$$z^{(i+1)} = \sigma\left( W_u \begin{bmatrix} u^{(i)} \\ z^{(i)} \\ x \end{bmatrix} + \begin{bmatrix} 2\alpha Vb \\ 2\alpha Vb \\ 0 \end{bmatrix} \right)$$

Here, $\sigma$ applies only to the top $h$ coordinates corresponding to $z^{(i)}$. Finally, we multiply the output of the monDEQ after $k$ iterations with the output weights, to obtain $f_o(x)$ as

$$f_0(x) = W_f \begin{bmatrix} u^{(k)} \\ z^{(k)} \\ x \end{bmatrix} + b_o$$

Thus, the equivalent depth-$k$ DNN which we construct would have weight matrices $W_1, [W_u]_{i=1}^{k-1}, W_f$, and the ReLUs in the intermediate layers would only apply to the top coordinates corresponding to $z$.

## E  Generalization bound for unrolled monDEQ

In this section, we derive a generalization bound for an unrolled monDEQ after unrolling the forward-backward iterations $d$ times and constructing the equivalent depth-$d$ network. The analysis follows the derivation of the generalization bound for DNNs by (Neyshabur et al., 2018), but we have to be careful since the weight matrices across the hidden layers are all the same and are of a certain parameterized form. Since the analysis in (Neyshabur et al., 2018) does not include biases in the DNNs, here, we consider unrolling monDEQs which do not have the bias term $b$.

Take each layer weights to be

$$W_u = \begin{bmatrix} I - \alpha(I - W) & \alpha U \\ 0 & I \end{bmatrix}$$

where the network is now a function of

$$\bar{z}_i := \begin{bmatrix} z^{(i)} \\ x \end{bmatrix}$$

and the ReLU applies only to the $z^{(i)}$ component. And

$$W_f = \begin{bmatrix} W_o \\ 0 \end{bmatrix}$$

Our prior distribution will now be over $\Delta_{W_o}, \Delta_A, \Delta_B, \Delta_U \sim \mathcal{N}(0, \sigma^2 I)$. Similar to the corresponding step in C above, we have that with probability $\geq \frac{1}{2}$, the perturbations are each bounded by $\omega := \sigma \sqrt{2h \ln(16h)}$. Also

$$\|\Delta_W\|_2 = \|\bar{A}^T \bar{A} - A^T A + \bar{B} - B - \bar{B}^T + B^T\|_2$$
$$\leq \|\Delta_A\|_2^2 + 2\|\Delta_A\|_2 \|A\|_2 + 2\|\Delta_B\|_2$$

And so

$$\|\Delta_{W_u}\|_2 = \left\| \begin{bmatrix} \alpha \Delta_W & \alpha \Delta_U \\ 0 & 0 \end{bmatrix} \right\|_2$$
$$\leq \alpha \left( \|\Delta_A\|_2^2 + 2\|\Delta_A\|_2 \|A\|_2 + 2\|\Delta_B\|_2 + \|\Delta_U\|_2 \right)$$

We take $\beta = \max\{\|A\|_2, \|B\|_2, \|W_u\|_2, \|W_o\|_2\}$. We assume that $|\tilde{\beta} - \beta| \leq \frac{1}{d}\beta$ and $d \geq 2$, so we have $\beta < e\tilde{\beta}$. Then applying Lemma 2 in (Neyshabur et al., 2018), we have

$$|f_{W_u + \Delta_{W_u}} - f_{W_u}|_2 \leq eB\beta^{d-1}((d-1)\|\Delta_{W_u}\|_2 + \|\Delta_{W_o}\|_2)$$
$$\leq eB\beta^{d-1}((d-1)\alpha \left( \|\Delta_A\|_2^2 + 2\|\Delta_A\|_2 \|A\|_2 + 2\|\Delta_B\|_2 + \|\Delta_U\|_2 \right) + \|\Delta_{W_o}\|_2)$$
$$\leq eB\beta^{d-1}((d-1)\alpha(\omega^2 + 2\omega\beta + 3\omega) + \omega)$$
$$\leq eB\beta^{d-1}((d-1)\alpha(\frac{1}{d}\beta + 2\beta + 3) + 1)\omega$$
$$\leq e^2 B\tilde{\beta}^{d-1}((d-1)\alpha(\frac{e}{d}\tilde{\beta} + 2e\tilde{\beta} + 3) + 1)\sigma\sqrt{2h \ln(16h)} \leq \frac{\gamma}{4}$$

if we choose

$$\sigma = \frac{\gamma}{4e^2 B\tilde{\beta}^{d-1}((d-1)\alpha(\frac{e}{d}\tilde{\beta} + 2e\tilde{\beta} + 3) + 1)\sqrt{2h \ln(16h)}}.$$

Let

$$\sum \|W.\|_F^2 = \|A\|_F^2 + \|B\|_F^2 + \|U\|_F^2 + \|W_o\|_F^2$$

Then

$$\mathrm{KL}(A + \Delta_A, B + \Delta_B, U + \Delta_U, W_o + \Delta_{W_o}) \| P) \leq \frac{\sum |A, B, U, W_o|_F^2}{2\sigma^2}$$
$$= \frac{16e^4 B^2 \tilde{\beta}^{2d-2}((d-1)\alpha(\frac{e}{d}\tilde{\beta} + 2e\tilde{\beta} + 3) + 1)^2 2h \ln(16h)}{\gamma^2} \sum \|W.\|_F^2$$
$$\leq \frac{16e^2 B^2 \beta^{2d-2}((d-1)\alpha(\frac{1}{d}\beta + 2\beta + 3) + 1)^2 2h \ln(16h)}{\gamma^2} \sum \|W.\|_F^2$$

Now, instantiating Lemma 1 above, we have that with probability at least $1 - \delta$,

$$L_0(f) \leq \hat{L}_\gamma(f) + 4\sqrt{\frac{16e^2 B^2 \beta^{2d-2}((d-1)\alpha(\frac{1}{d}\beta + 2\beta + 3) + 1)^2 2h \ln(16h) \sum \|W.\|_F^2}{\gamma^2(M-1)} + \frac{\ln(\frac{6M}{\delta})}{M-1}}$$

Following Neyshabur et al. (2018), we need to take a union bound over $\tilde{\beta}$ so that the above result holds for all $\beta$. Observe that we only need to consider $\beta$ in the range

$$\left( \frac{\gamma}{2B} \right)^{1/d} \leq \beta \leq \left( \frac{\gamma \sqrt{M}}{2B} \right)^{1/d}.$$

If $\beta^d \leq \frac{\gamma m}{2B}$ then $|f(x)| \leq \beta^d B \leq \frac{\gamma}{2}$ so $\hat{L}_\gamma(f) > 1$. If $\beta^d \geq \frac{\gamma \sqrt{M}}{2B}$ then the second term in the first term in the numerator is greater than 1 so the result holds trivially. So $|\beta - \tilde{\beta}| \leq \frac{1}{d} \left( \frac{\gamma}{2B} \right)^{1/d}$ guarantees $|\beta - \tilde{\beta}| \leq \frac{1}{d}\beta$ in this range. So we can use a cover of size $dM^{1/2d}$, just as in Neyshabur et al. (2018).

## F    ADDITIONAL MONDEQ LIPSCHITZ BOUND RESULTS

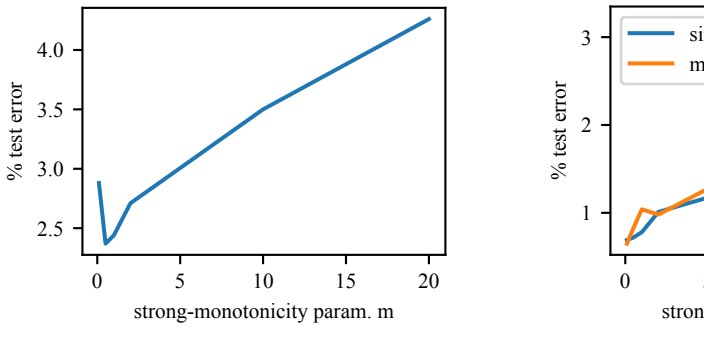

(a) Width 40 monDEQs shown in Figure 1b    (b) Convolutional monDEQs shown in Figure 1d

Figure 5: Test error for the monDEQs from Section 5.1

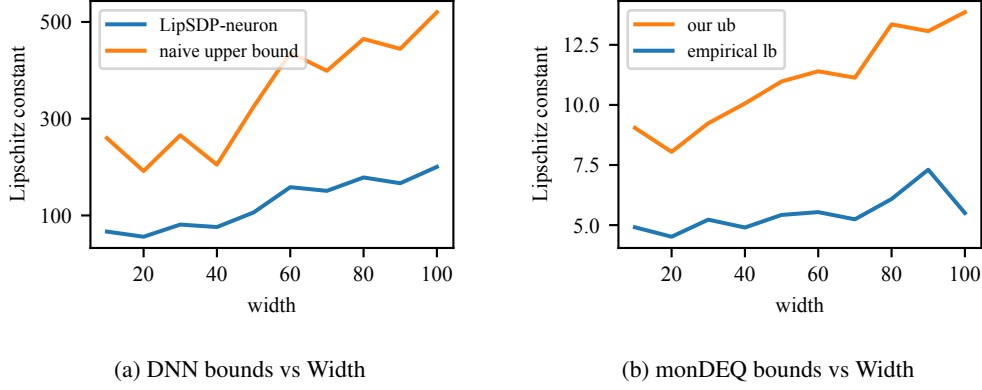

(a) DNN bounds vs Width    (b) monDEQ bounds vs Width

Figure 6: Evaluating Lipschitz bounds as a function of width. lb: lower bound; ub: upper bound.

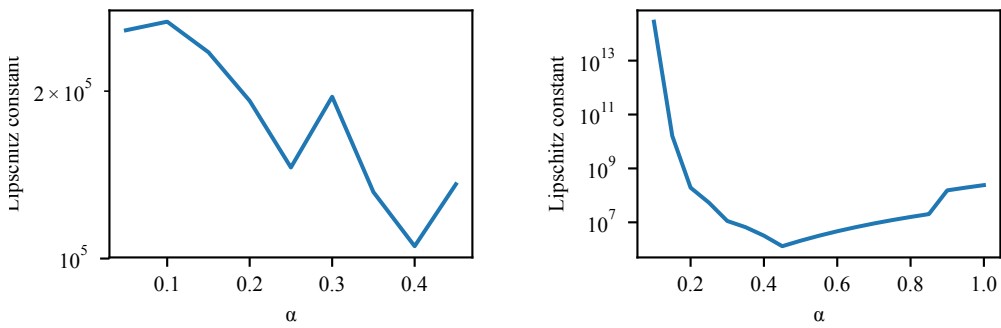

(a) Forward-Backward unrolled Lipschitz bounds    (b) Peaceman-Rachford unrolled Lipschitz bounds

Figure 7: Lipschitz bounds for monDEQ by unrolling forward-backward or Peaceman-Rachford, for a range of $\alpha$.

**MNIST test error** In Figure 5 we plot the test error of the width 40 monDEQs for which the Lipschitz bounds are given in Figure 1b. We see that it increases from 2.4% for $m = 0.5$ to 4.2% for $m = 20$. For comparison, the DNNs shown in Figure 1a have test error between 2.8% and 3.2%, with no trend w.r.t. depth.

**MNIST width experiments** Figure 6 shows the lower and upper bounds for DNNs and monDEQs of varying widths, as described in Section 5.1. Figure 6a is transcribed from Fazlyab et al. (2019) figure 2(a).

**CIFAR-10 CNN models** The details of the CNN sm, CNN med, and CNN lg models used in Section 5.1 are as follows:

CNN sm:

| Layer # | Layer | # channels out | stride |
|---------|-------|----------------|--------|
| 1 | Conv2D | 32 | 1 |
| 2 | Conv2D | 32 | 2 |
| 3 | Linear | 100 | |
| 4 | Linear | 10 | |

CNN med:

| Layer # | Layer | # channels out | stride |
|---------|-------|----------------|--------|
| 1 | Conv2D | 32 | 1 |
| 2 | Conv2D | 32 | 2 |
| 3 | Conv2D | 64 | 1 |
| 4 | Conv2D | 64 | 2 |
| 5 | Linear | 100 | |
| 6 | Linear | 10 | |

CNN lg:

| Layer # | Layer | # channels out | stride |
|---------|-------|----------------|--------|
| 1 | Conv2D | 32 | 1 |
| 2 | Conv2D | 32 | 2 |
| 3 | Conv2D | 64 | 1 |
| 4 | Conv2D | 64 | 2 |
| 5 | Conv2D | 128 | 1 |
| 6 | Conv2D | 128 | 2 |
| 7 | Linear | 100 | |
| 8 | Linear | 10 | |

All models use ReLU activations and kernels of size 3.

**Unrolling monDEQs** The approximate upper bounds on monDEQ Lipschitz constants obtained by unrolling the operator splitting methods are shown in Figure 7.

## G ADDITIONAL MONDEQ GENERALIZATION BOUND RESULTS

We validate that both the monDEQ and DNN models considered in Section 5.2 for generalization bound comparisons indeed achieve comparable accuracy as well. Figures 8a, 8b illustrate this: we can observe that the various models considered all achieve test error in the same range.

Next, we compute generalization bounds for unrolled monDEQs (Section E) for a variety of $\alpha$ values, to get a sense of the quality of our generalization bound for the monDEQ. In Figure 8c, we can observe that the generalization bounds for these unrolled networks are larger (on the order of $10^6$) as compared to the generalization bound for the monDEQ (on the order $10^5$). This shows that our bound is tighter.

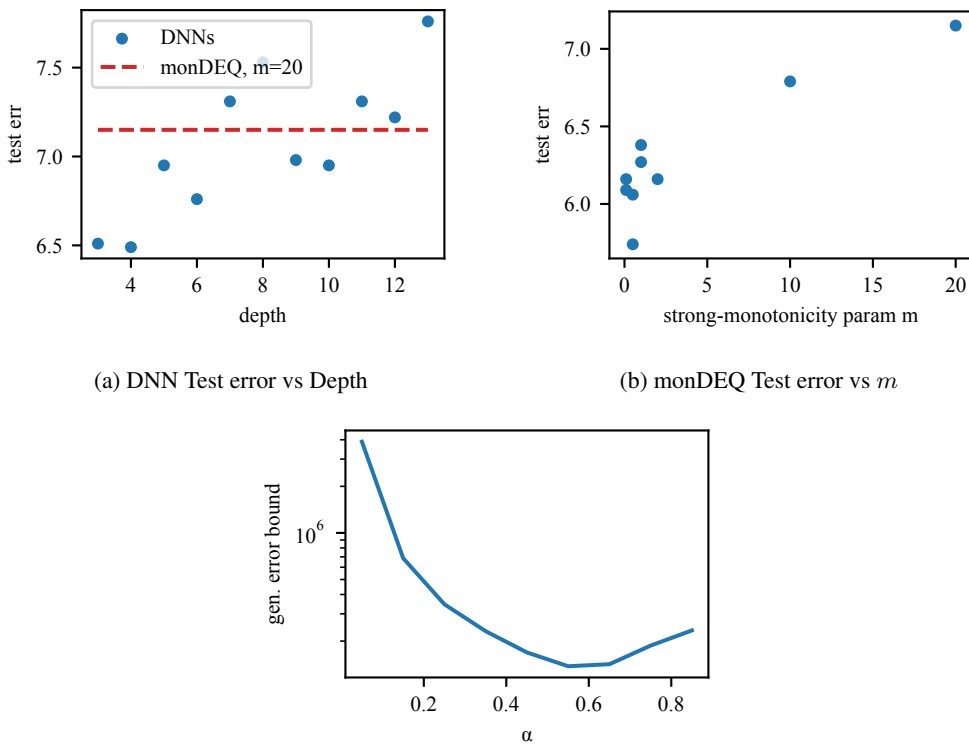

(a) DNN Test error vs Depth

(b) monDEQ Test error vs $m$

(c) Forward-Backward unrolled generalization bound

Figure 8: Test error for DNNs and monDEQs, and monDEQ generalization bounds by unrolling forward-backward iterations.

## H    ADDITIONAL ADVERSARIAL ROBUSTNESS RESULTS

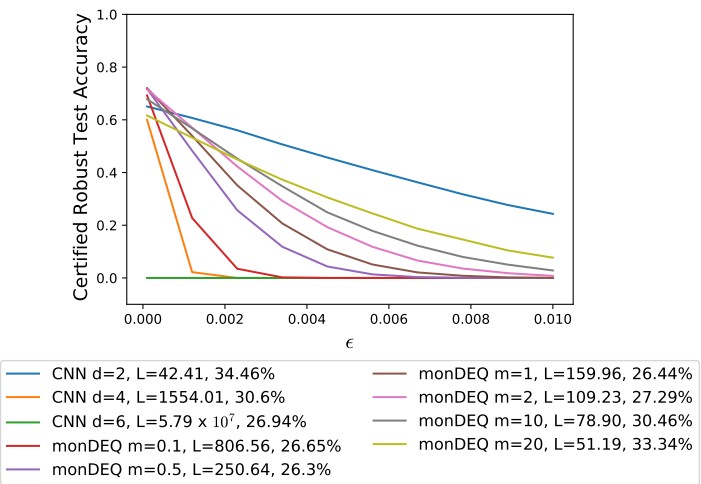

(a) Certified Adversarial Robustness - In the legend, we state the Lipschitz constant of each model, and its (clean) test error

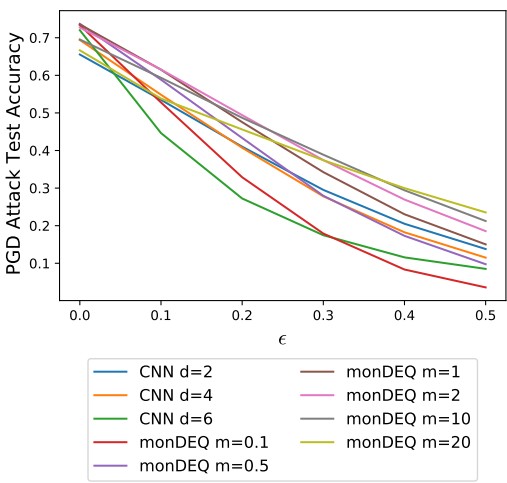

(b) Adversarial robustness to attacks

Figure 9: Adversarial robustness of monDEQs as compared to CNNs

We run the same experiments that we ran in Section 5.3 on the CIFAR-10 dataset. We train single convolution monDEQs (128 channels) with a range of $m$ values, as well as 3 CNN models: CNN sm, CNN med and CNN lg whose architectures are described in Appendix F above.

As in the case of the MNIST experiments, we see that varying $m$ gives a range of trade offs between clean accuracy and adversarial robustness. In terms of certified robustness, we see in Figure 9a that all of the monDEQ s besides $m = 20$ have both better clean *and* better robust accuracy than all the CNNs (with the exception of the CNN with $d = 2$, which has much lower clean accuracy). In Figure 9b, in terms of empirical robustness, we see that $m$ parameterizes a similar trade-off between robust and clean accuracy. In fact, it is possible to choose $m$ (e.g. $m = 1$ here), such that the monDEQ outperforms all CNNs in terms of clean accuracy and at all attack sizes.

