# OpenReview forum: "Estimating Lipschitz constants of monotone deep equilibrium models"
_ICLR.cc/2021/Conference — ICLR 2021 Poster_

### Official Review · AnonReviewer2 · 2020-10-26
**Interesting direction, but lacks clarity and potential impact**

**Rating:** 6
**Confidence:** 5

**Review:**

#### Summary
The authors of this paper derive analytic expressions for the Lipschitz constants (LCs) of monotone deep equilibrium networks (mDEQs) with respect to their inputs and learnable parameters.  Further, they leverage these estimates to provide a PAC-Bayes style generalization bound.  Finally, they empirically calculate their bound and compare to relevant approaches for computing the LCs of feed-forward networks.

#### Strengths
- The simplicity of the results is both a strength and (as I will discuss later) something of a weakness.  On the positive side, the authors show that their bounds do not have an exponential dependence on the depth of the network (which is especially important given that these networks are in some sense "infinite-depth" neural networks).  I checked each of the proofs, and they all appear to be correct.
- The ability to control (an upper bound on) the LC with the strong monotonicity parameter $m$ is quite useful here.  Indeed, there is no such term in any of the methods that can be used to derive an upper bound on the LC of a feed-forward NN.
- The PAC-Bayes style bound is interesting, and should be further examined in future work.

#### Weaknesses
- It's not immediately clear to me why this problem is of particular use to the broader community.  Much of the literature that seeks to derive LCs for feed-forward/convolutional NNs is motivated by the connections between adversarial robustness and the LC (see, for example, Prop. 1 in (Fazlyab et al., 2020)) or by the prevalence of LCs in generalization bounds.  However, as far as I am aware, it is not known whether mDEQs are fragile to adversarial examples.  Furthermore, as the mDEQ network is such a new architecture (the paper came out earlier this year in 2020), it remains to be seen whether this architecture will be broadly and/or practically useful for the deep learning community, whereas it has been shown repeatedly that feed-forward networks and CNNs are broadly applicable in a variety of domains (see e.g. LeCun et al., 2015).  So while this problem may be interesting from a technical point of view, I am not sure whether it is a useful result that will have noteworthy down-the-line implications.
- The technical results of Props 1-2 and Theorms 1-2 are quite straightforward, and as far as I can tell, the argument used to show Theorem 3 is quite similar to that of Neyshabur et al (2018).  Compared to works such as Fazylab et al (2019) and Raghunathan et al (2018), both of which build novel semidefinite programming frameworks, and Latorre et al (2019), which exploits underlying structure to derivate a polynomial optimization scheme, there is significantly less technical innovation here, and no new tools are developed that might be more broadly useful to the community.
- The presentation of the experiments is sloppy.  Empirical bounds are computed for mDEQs, but not for other architectures.  Further, abbreviations (e.g. lb == lower bound?) are used which are not defined in the text.  Furthermore, given the apparent scalability of this approach, I would argue that performing experiments only on MNIST is not enough.  Why not try to compute bounds for more challenging/higher-dimensional datasets?

#### Literature review
- The literature review concerning the work that has been done toward deriving upper bounds on the LC of DNNs is relatively thorough.  The bounds in this paper are the first that I am aware of that calculate LCs for mDEQs, so there are no competing methods here.

#### Points of clarification and questions
- Is there any way to extend you analysis to other norms?  Given that Theorems 1 and 2 essentially follow from applications of Cauchy-Schwarz and the triangle inequality, it seems likely that more general results hold for other p-norms.
- As written, the proofs of Theorem 1 and Prop 2 rely on $\sigma$ being the ReLU activation function.  However, the argument holds more generally for 1-Lipschitz activations.  The authors should clarify this, as it is confusing to appeal to properties of ReLU when an easy and more general result holds.
- In the proof of Theorem 1, the authors gloss over some details that I feel should be more clearly explained.  For example, one must rely on _continuity_ of the $2$-norm $||\cdot||_2$ to exchange the limit with the norm.  This is used but not stated, and may confuse some readers.
- In the proof of Prop 2, an implicit assumption that $||T||_2 \leq L[T]$, where $L[T]$ denotes the Lipschitz constant of the operator $T$ (with respect to the 2-norm, although the norm of choice is actually not explicitly specified when defining this notation).  However, I'm not sure whether this is actually true.  Given that $T$ is applied linearly in this case, it certainly seems to hold that $L[T] \leq ||T||_2$, as $||Tx - T_y||_2 = ||T(x-y)||_2 \leq ||T||_2 ||x-y||_2$.  However, if the other direction holds (i.e. $||T||_2 \leq L[T]$), then given the previous sentence, it would follow that $||T||_2 = L[T]$, meaning that we could have an equality in that step of the proof rather than an inequality.  If I have misunderstood something here, clarifications by the author would be helpful.
- In the proof of the generalization bound, it is assumed that all perturbations are generated by a normal distribution with mean 0.  I wonder if the authors can say anything about perhaps the more interesting case in which perturbations are _adversarially chosen_ rather than generated via a known distribution.
- How were the empirical estimates of the LCs computed for the mDEQ bounds in the experiments (e.g. Figs 1b and 2b).  It seems that to offer a fair comparison, empirical estimates should also be computed for feed-forward DNNs.
- The authors mention that both mDEQs and feed-forward NNs are trained to obtain "similar test accuracies."  These accuracies should be reported for further clarity.
- It looks to me that Figure 2(a) is incorrectly transcribed from Fazlyab et al. (2019).  The authors should double check this.
- What are the "expected margin loss" and "empirical margin loss"?  These are used in the PAC bound, but I don't see the definition of these losses.

#### Final thoughts
The problem studied here is certainly interesting and has not been studied before in the literature.   The proofs rely on straightforward mechanisms, making them easy to follow, and the bounds do not suffer from exponential dependence on network depth, which has been an issue for previous bounds.  Further, the literature review is solid and empirically the bounds seem to be low relative corresponding DNNs.  However, I am not sure whether this problem will be of interest to many others in the community, given that it is not yet clear how broadly applicable mDEQs are.  Further, I am not sure there are enough results here to merit a full paper.  There are a number of ways that the authors could further their analysis within the scope of this topic, including extending the analysis to include other norms.  Indeed, one could imagine leveraging the analysis via alpha-averaged operators of Combettes et al. (2019) to potentially refine these bounds.  Such connections could be explored and exploited, and room could be made in the main text by pushing the proofs to the appendix.  Finally, some of the writing and proofs are a little bit imprecise (as I discuss above).  For these reasons, I am leaning toward suggesting rejection for this paper, as I believe that with further refinement the authors could submit a more complete version to a future conference.

---

> ### Author Response · Authors · 2020-11-20
> **Response to AnonReviewer2**
>
> Please see our comments at the top regarding adversarial robustness and empirical lower bounds, and test error. We have incorporated a number of the suggested changes into the pdf, and include additional comments here:
>
> ---
>
> #### *"Why not try to compute bounds for more challenging/higher-dimensional datasets?"*
>
> We are experimenting with models on CIFAR10 and hope to include something before the end of the response period.
>
> ---
>
> #### *"Is there any way to extend your analysis to other norms?"*
>
> Thank you for pointing this out. We are looking into it, but it is not clear from our initial attempts that the arguments could be easily generalized to other norms. For example, in the proof steps of Proposition 1, we required expanding out $\|Tx-Ty\|_2^2$, which gave rise to an inner product which we could bound using the strong monotonicity condition on $I-W$. It is not clear how we would do this part of the calculation for say the $\ell_\infty$ norm.
>
> ---
>
> #### *"...the argument holds more generally for 1-Lipschitz activations"*
>
> Thank you for pointing this out. You are correct,  the proofs go through more generally for any activation function which satisfies the conditions of Winston & Kolter 2020, that is, $\sigma = \mathrm{prox}_f^1$ for some convex, closed  proper function $f$.
>
> ---
>
> #### *"In the proof of Prop 2, an implicit assumption that $\|T\|_2\leq L[T]$"*
>
> There is a typo in the 4th step in the proof of Proposition 2.  The $\leq$ should instead be an $=$ sign. You are right, since $T$ is a linear operator, $L[T] = |T|_2$. We have fixed the typo in the proof.

---

> > ### Comment · AnonReviewer2 · 2020-11-23
> > **New experiments are solid; would still like to see higher-dimensional datasets**
> >
> > > Re: "Empirical lower bounds on Lipschitz constants" and "Test error of monDEQs vs standard NNs"
> >
> > This looks great.  Thanks for clarifying these points.
> >
> > ___
> >
> > > "We are experimenting with models on CIFAR10 and hope to include something before the end of the response period."
> >
> > Great -- I'm looking forward to seeing these results.
> >
> > ___
> >
> > > Re: "Adversarial robustness"
> >
> > I've looked over this new experiment.  I think this experiment is necessary and is an important addition to this work.  Based on the results, it seems that depending on the choice of $m$, mDEQs can be more or less robust than DNNs.  Specifically, it seems that for $m=0.1$, mDEQs are less robust than DNNs, and for $m=20$, they are more robust.
> >
> > It would have been quite nice if for more values of $m$, mDEQs were shown to be more robust than DNNs, but it seems that that is not true in general.  In any case, this is not a weakness of the current paper.
> >
> > ___
> >
> > > Overall rebuttal response
> >
> > The extra post-rebuttal experiments seem strong and the clarifications addressed several of my points.  I still have concerns about whether this paper will have high impact given that mDEQs are so new, and I agree with R3 that the techniques used are fairly standard.  Further, I would still like to see results on CIFAR-10 or other higher-dimensional datasets, as I still feel that MNIST is not enough.  With CIFAR experiments, I would lean toward increasing my score to a 6, given the other solid experiments that have been done post-rebuttal.

---

> > > ### Author Response · Authors · 2020-11-25
> > > **CIFAR-10 experiments added**
> > >
> > > Thanks very much for all your detailed feedback. The pdf has been updated to include CIFAR-10 experiments (Lipschitz bounds in Section 5.1 and adversarial robustness in Appendix I).

---

### Official Review · AnonReviewer4 · 2020-10-27
**User friendly upper bounds on the Lipschitz constants of monotone DEQs**

**Rating:** 6
**Confidence:** 3

**Review:**

This work analyses the Lipschitz constant of monotone Deep Equlibrium models by exploiting the fixed point formulation of the input/output relation. They derive analytical upper bounds to the Lipschitz constants both with respect to input perturbations (L_in) as well as with respect to the network's weights perturbations (L_w). These bounds depend on the networks parameters and do not involve exponential constant, as opposed to Deep Neural Networks that suffer from their Lipschitz constant being exponential in the depth.

In particular, the derived upper bound for L_in oinly depends on the strong monotonicity parameter m and the input injection weight matrix U (and not on the hidden unit weight matrix W). These easily computable upper bounds allow for controlling the Lipschitz constants by carefully selecting the weight parametrization, and especially the strong monotonicity parameter m. This is useful for ensuring robustness of the model with respect to adversarial perturbations.

It is well known that bounds on L_w can be used to derive generalisation bound. By using their bound on L_w, the authors show a new generalisation bound for trained monDEQ models, with polynomial dependence on the network parameters.

1. I am not so familiar with DEQ's but I imagine that the input injection weight matrix U is also learned during training? In this case, setting a value of m is not sufficient for controlling the L_in, since the upper bound also depends on \|U\|_2.

2. In a similar way that the finite depth of a DNN limits its capacity, I believe that the strong monotonicity parameter of a DEQ also limits its capacity. For a fair comparison between the dependence of each model on depth/m, it would be nice to compare the dependence on depth/m on the capacity of respective models.

Figure 4a: The value of the test error for the dashed red line (about 6.7) does not correspond to the value of the test error for monDEQ with m=20 as shown in Figure 4b (about 7.1)

---

> ### Author Response · Authors · 2020-11-20
> **Response to AnonReviewer4**
>
> #### *”...setting a value of $m$ is not sufficient for controlling the $L_{in}$, since the upper bound also depends on $|U|_2$”*
>
> This is a good point and we will be sure to clarify this in the paper. The Lipschitz constant does indeed depend on the spectral norm of the trained matrix $U$. We simply wanted to emphasize that we can have a non-trivial handle on the Lipschitz constant of the monDEQ via the dependence on $m$, which is set at design-time, and which empirically does  demonstrate a strong influence on $L_{in}$.
>
> ---
>
> #### *“....it  would be nice to compare the dependence on depth/$m$ on the capacity ”.*
>
> Thank you for this suggestion. Our motivation for including the figure in Appendix G showing DNN and monDEQ test error was precisely this - to illustrate the similar capacity of the DNNs and monDEQs we considered. However, the existing plots were corresponding to models considered in Section 5.2 on generalization bounds. To be more comprehensive, in Appendix H we have additionally included a discussion of the test error of the DNNs and monDEQs used in Section 5.1. There the monDEQs show a similar trend with $m$. However, in contrast, the DNNs used in Section 5.1 do not show increasing test error with depth. We believe this  is because they were not trained to reach a particular margin.
>
> ---
>
> ####  *Figure 4a: The value of the test error for the dashed red line (about 6.7) does not correspond to the value of the test error for monDEQ with m=20 as shown in Figure 4b (about 7.1).*
>
> Good catch. This was due to a discrepancy in training which we will correct in the final version of the paper. The DNNs and MON with m=20 in figure 4a (now figure 5a)  were trained with a particular weight initialization scheme, while those in figure 4b were trained with a different initialization.

---

### Official Review · AnonReviewer1 · 2020-10-28
**A stepping stone towards understanding the generalization and robustness properties of Deep Equilibrium models. Some comparison to relevant work is missing, which is a major issue.**

**Rating:** 7
**Confidence:** 4

**Review:**

**after rebuttal**: The authors have addressed some of my major concerns in an updated version. for this reason I raise my score to a point where I can recommend acceptance. I now add after-rebuttal comments at the end of each item of my original review.

## Summary
This work obtains upper bounds on the Lipschitz constant of a Monotone Deep Equilibrium Model (monotone DEQs) depending only on their strong monotonicity parameter $m$. This is in contrast to the naive bound for deep neural networks, which degrade with the depth. This also implies that controlling the smoothness of a monotone DEQ is possible just via the single parameter $m$, rather than controlling operator norms of each layer of a DNN. Additionaly, the authors derive generalization bounds  for such type of models, based on the deterministic PAC-bayes approach. This generalization bound reveal a dependency on the strong monotonicity parameter $m$, as well as the size of the hidden layer and appears to be the first result of such kind.

## Pros:
**1. Clarity: The main claims of the paper and their exposition are clearly stated**: The main contributions i.e., the upper bound on Lipschitz constant (Thm1) Upper bound on change after perturbation of the weights (Thm2) and Generalization bound (Thm3) draw from well developed techniques from monotone operator theory and statistical learning theory, using well known concepts which are accessible to researchers with basic knowledge in such areas.

**2. Significance: It appears that DEQs have many advantages like reduce memory usage as well as good performance, hence it is of major importance to understand their smoothness properties and generalization guarantees, which this work contributes to**. Controlling the smoothness of traditional neural networks seems to suffer from a computation-quality tradeoff where simple bounds on the Lipschitz constant are easy to enforce, but are of dubious quality, while tight estimates are computationally inefficient. This work provides evidence that exerting such control on monotone DEQs is conceptually easier.

**3. Originality: although the results mostly come from standard techniques, they are useful and novel, to my knowledge**. Previous work has mostly focused on computational aspects and variations, illustrating the feasibility of the approach.

## Cons:
**1. Originality/Significance: there is a major relevant work that is not cited nor compared to.** There is some work studying bounds on the Lipschitz constant  of implicit models, although more of the flavor of Neural ODEs https://arxiv.org/pdf/2004.13135.pdf
this work also appears to focus on the Lipschitz constant w.r.t. the parameters of the networks. I have to accept that the results there are somewhat convolved but I think this work should be cited and the differences between this work should be clarified

There is also the following work https://arxiv.org/abs/1908.06315v4 (note that the v4 is recent so it might not classify as prior work given that the deadline was beginning of october, but there is an initial version v1 dating from 2019). THis work also studies many aspects of implicit models. In particular it looks like section 4 in v4 deals with Lipschitz constants with respect to the L-infinity norm, which is related to the current work but it is not cited. Again I think this work should be cited and the differences with the submitted work should be clarified. **after rebuttal**: The authors have included such references and a discussion of the main differences (end of section 2), making clear how their approach differs.

**2. Clarity: Some claims are misleading, regarding the Lipschitz constant**: It is claimed that theorem 1 shows that the Lipschitz constant does not depend on the matrix W. However the distinction should be made that the value obtained is an upper bound, so in fact it is possible that the minimal Lipschitz constant depends on W, but this particular upper bound obtained does not. This should make the claim more clear. **after rebuttal**: authors acknowledge this issue, for some reason I don't see that this is fixed in the new version, but could be fixed with minor rewriting in the final version. They should only write the conclusions for their **derived upper bound on the Lipschitz constant** rather than **THE Lipschitz constant** which is traditionally understood as being an infimum over the set of possible constants with the Lipschitz property.

**3. Clarity: It looks like the proposition 2 is not used anywhere and does not correspond to any substantial claim and thus should be removed**. Unless I am missing something, I don't understand why is preposition 2 relevant or how it is used to support the important theorems. After checking, it is not used in the proofs of the theorems. Am I missing something? **after rebuttal** the authors do not seem to address this. I have now realized that Proposition 2 is used in the proof of theorem 2 but because such proof is found in the appendix, it seems that including an intermediate result in the main text is a poor stylistic choice. However this is not a major issue.

**4. Significance: I think that the pros of the generalization bound (no explicit dependence of depth) is great, but its weaknessess are downplayed. In particular the generalization bound depends linearly in the width $h$**. Weaknessess should also be acknowledged. In contrast, as far as I know the DNN gen bounds like that of Bartlett et al. or Neyshabur et al. depend only logarithmically on the width.

## Other comments:
1. THeorem 3: typo, change m to M(set of size M)
2. In the experiments in 5 it is weird to use the "lower bound" from Combettes et al. as it is not really a lower bound. I don't remember what is the motivation to use this but seems really confusing to use something which is not a lower bound and call it lower bound. A better lower bound could be sampling points and taking the maximum norm of the gradients as is done in other papers. Could this be changed/added easily? **after-rebuttal**: the authors have changed the lower bound to a true lower bound.

All in all, I think the missing references/discussion are a major point that has to be addressed. Hopefully this can be done in moderate space/time during rebuttal. In that case I would be willing to increase my score because my overall impression is positive.

---

> ### Author Response · Authors · 2020-11-20
> **Response to AnonReviewer1**
>
> Please see our comments at the top regarding empirical lower bounds.
>
> ---
>
> Thank you for pointing out these relevant works. We will definitely include a discussion of these in the final version of the paper. Here, we briefly provide some comparison:
>
> - [Implicit deep learning](https://arxiv.org/abs/1908.06315v4): They define restrictive conditions for well-posedness of an implicit network which are different from those of the monDEQ. In essence, they require the weight matrix $W$ to be such that forward iteration is stable (as opposed to the stability of the operator splitting methods required by monDEQ). They derive Lipschitz constants and robustness guarantees under these conditions; for example when $\|W\|_{\infty} < 1$, then a Lipschitz bound can be derived by simply manipulating the fixed-point equation as they demonstrate in equation 4.3.
>
> - [Estimating Full Lipschitz Constants of Deep Neural Networks](https://arxiv.org/pdf/2004.13135.pdf): The framework they define in Section 4, incorporates the Neural ODE (which is the solution of an ODE at a given time T) but not the monDEQ (which can be cast as finding the *equilibrium point* of an ODE). Thus, it does not appear their framework could be directly applied to estimate the Lipschitz constant of the monDEQ layer. Furthermore, their bounds for traditional DNNs (Corollary 3.3) involve a $B^u$ term where B is the max weight norm and $u$ is network depth. This exponential dependence on depth for DNNs is similar to that derived in Lemma 2 in Neyshabur et al. (2018).
>
> ---
>
> #### *”Clarity: Some claims are misleading, regarding the Lipschitz constant”*
>
> Thank you for pointing this out. We will explicitly clarify this in the body of the paper. We agree that the minimal Lipschitz constant might indeed depend on $W$, even if our upper bound does not. However, we would like to point out that in our experiments, we observe that our bounds are relatively tight (in the sense that they are close to the lower bounds), so the true Lipschitz constant can’t depend on  $W$ *very much.*
>
> ---
>
> #### *”Clarity: It looks like the proposition 2 is not used anywhere”*
>
> Proposition 2 is used in the Proof of Theorem 2 which is outlined in Appendix C. Based on your feedback, we will move the statement of the proposition too to the Appendix.
>
> ---
>
> #### *”...the generalization bound depends linearly in the width $h$”*
>
> We would like to note here that the generalization bound for DNNs derived in Neyshabur et al (2018) also has the same dependence on $h$ as our bound which is $\sqrt{h ln(h)}$. A detailed comparison with the bound in Bartlett et al (2017) is provided in Section 3 of Neyshabur et al (2018), and we acknowledge that the bound in Bartlett et al (2017) can have a better dependence on $h$ in the cases they have outlined. We will mention this explicitly in the paper.

---

> > ### Comment · AnonReviewer1 · 2020-11-24
> > **when will the comparison be included in the paper?**
> >
> > Including these references and comparisons will definitely improve the paper and it should be done before the next reviewing phase (Nov 24.)

---

> > > ### Author Response · Authors · 2020-11-25
> > > **References and comparisons added**
> > >
> > > The pdf has been updated to include this at the end of Section 2. Thanks again for pointing out these missing comparisons.

---

### Official Review · AnonReviewer3 · 2020-10-28
**Review: Interesting paper but more impact is needed**

**Rating:** 5
**Confidence:** 4

**Review:**

The authors consider the problem of estimating the Lipschitz constant of a specific type of network known as a Monotone Deep Equilibrium Model (monDEQ). These models are a recursive model designed such that the recurrence equations converge to a fixed point. Previous approaches to bounding the Lipschitz condition work on fixed depth DNNs and would not work in this case, as monDEQs are essentially infinite length. The authors develop an approach to bound the Lipschitz constant on monDEQs, both when we consider the model a function of the input and a function of the variables.  They then use their bounds to establish generalization results for monDEQs. Finally they use their bounds to experimentally compare monDEQs to DNNs in terms of their Lipschitz constants.

While I think the results in this paper are interesting, I think that they are somewhat incremental. The approaches used are fairly standard techniques. Additionally, the ML field is not yet broadly interested in monDEQs. The authors argue that monDEQs are powerful due to their lower Lipschitz constants compared to DNNs which still achieving good statistical results. They can strengthen the paper by explicitly demonstrating that this is indeed true, say by training networks with good statistical performance compared to DNNs that are more resistant to adversarial attacks.

The theory sections are well written. I think adding more details to the captions can improve the clarity of the experimental section.

The authors write "We note that the models obtain similar test accuracy to the DNNs." Can we get a more quantitative comparision of the statistical performance of monDEQs to DNNs?

"we aim to analyze the vacuousness of the derived generalization bound" -> Can you clarify this statement? What is the vacuousness of the derived generalization bound?

Overall, while I feel the paper is interesting, I feel it needs more impact before it is ready for acceptance. Further justifying the user of monDEQs based on their Lipschitz properties can increase the impact of the paper.

---

> ### Author Response · Authors · 2020-11-20
> **Response to AnonReviewer3**
>
> Please see our comments at the top regarding adversarial robustness and test error of monDEQs.
>
> ---
>
> #### *"we aim to analyze the vacuousness of the derived generalization bound"*
>
> By vacuousness of our bound, we mean that the bound is (much) larger than 1. While the generalization error is of course always less than 1, most existing generalization bounds for DNNs are in fact several orders of magnitude too large (refer to Figure 2(a) in https://arxiv.org/pdf/1905.13344.pdf). However, while existing generalization bounds for DNNs typically have exponential dependence on depth, our monDEQ bound doesn’t, which is why we feel it might be interesting to further analyze our bound.

---

### Official Review · AnonReviewer5 · 2020-11-04
**Official Blind Review #5**

**Rating:** 5
**Confidence:** 4

**Review:**

This paper analyzes the Lipschitz constant of a recently proposed implicit-depth model, the monotone deep equilibrium model (monDEQ). Simple-form bounds on the Lipschitz constant w.r.t both the inputs and the weights are derived. The derived bounds are used to show an attracting property of the monDEQ model that its Lipschitz bound and PAC-Bayes generalization bound does not have exponential dependence on network depth as normal DNNs.

Pros:
1.	The paper is well-written and easy to read.
2.	The proofs for Lipschitz bounds that I have checked are correct.

Cons:
1.	My major concern is the rather limited contributions of this paper. The methods for estimating the Lipschitz bound and the PAC-Bayes generalization bound are not novel. And the result that the Lipschitz constant of the monDEQ model is depth-independent is interesting but not surprising, since in some sense the DEQ model could be viewed as only having one layer.
2.	When deriving the Lipschitz bound, I don’t understand why to take the limit as $\alpha → 0$ since $\alpha$ is also a hyperparameter. In the experiment part, which $\alpha$ is using for each model? And if the bound involving $\alpha$ is used, will it be much tighter?
3.	The experiment regarding the Lipschitz constants of the unrolled monDEQs is confusing to me. I don’t see why the Lipschitz constant computed in this way is much larger than the derived bound. Instead, I suppose it should be close to the bound because in the end it would also be computed as a convergent series similar to the derivation in the paper. Could you explain this in more detail? Besides, the notations and derivations in Appendix E are not very clear.
4.	In the experiment of the generalization bound, why does the computed bound increase as $m$ increases (Fig. 3(b)), but the test error increases (Fig. 4(b))?

Additional Comments/Questions:
1.	In the experiments, how is the empirical lower bound of the monDEQ Lipschitz constant computed?
2.	Typos in the related work part: “… several orders of magnitude larger *than than* empirical lower bounds …” and “The DEQ model directly solves *for for* the fixed-point …”
3.	Typo in Theorem 3: “training set of size $m$”-> “training set of size $M$”

---

> ### Author Response · Authors · 2020-11-20
> **Response to AnonReviewer5**
>
> Please see our comments at the top regarding empirical lower bounds.
>
> ---
>
> #### *“When deriving the Lipschitz bound, I don’t understand why to take the limit as $\alpha \rightarrow \infty$  since $\alpha$  is also a hyperparameter…”*
>
> The output of the monDEQ is simply the fixed point of the equation defined in (2), and does not depend on $\alpha$. Thus, the Lipschitz constant of the monDEQ does not depend on $\alpha$ either. $\alpha$ is simply a parameter of the forward-backward iterations, which affects the convergence rate of the forward-backward iterations but not the point they converge to. In the experiments we chose an $\alpha$ small enough to ensure that the forward-backward iterations converged within a tolerance of 1e-5 (the iterations are guaranteed to converge provided $\alpha$ is small enough), but different choices of $\alpha$ would have still converged to the same fixed point. In the proof of Theorems 1 and 2, we unrolled the forward-backward iterations only as a *tool* for computing the Lipschitz constant of the monDEQ. Then, because the limit as $k\rightarrow\infty$ does not depend on $\alpha$, we get a bound that holds for all $\alpha$, and which is minimized by taking the limit $\alpha\rightarrow 0$.
>
> ---
>
> #### *“The experiment regarding the Lipschitz constants of the unrolled monDEQ….”*
>
> Apologies for the unclear explanation. We have made Appendix E more elaborate. In these experiments, we are effectively constructing an equivalent depth-$d$ feedforward network that computes the map given by iterating forward-backward/peaceman-rachford iterations $d$ times. Since there is input-injection involved in the monDEQ layer, to achieve the same map we have to necessarily construct a block matrix which includes an identity block, so it has spectral norm greater than $1$ unlike $I-\alpha(I-W))$. Computing the naive upper bound on the Lipschitz constant of this neural network amounts to multiplying the spectral norm of the block matrix $d$ times. Since we are exponentiating the spectral norm of this block matrix with spectral norm greater than $1$, it turns out to be much larger than the bounds we derived for the equivalent monDEQ.
>
> ---
>
> #### *“In the experiment of the generalization bound, why does the computed bound increase as $m$ increases, but the test error increases?”*
>
> We agree with the observation made by the reviewer that the generalization bound indeed decreases as $m$ increases, but the test error increases. Although this observation seems counter-intuitive, we would like to point out that the test error need not be strictly positively correlated with the value of the generalization bound. By that, we mean that a model that achieves very large test error could also have a small value of the generalization bound.

---

> > ### Comment · AnonReviewer5 · 2020-11-24
> > **Nice experiments on adversarial robustness, but the contribution is still the biggest concern**
> >
> > Thanks for your clarifications! I really like the experiments on adversarial robustness which make the paper more solid. But my biggest concern is still the contribution and potential impact of the paper, since the methodology isn't novel and monDEQ hasn't attracted many others in the community yet. If authors could elaborate more about this, I would tend to raise my score to 6.

---

> > > ### Author Response · Authors · 2020-11-25
> > > **Addressing potential impact**
> > >
> > > Thank you, we can appreciate that concern -- it's true that monDEQ has not gained widespread attention yet. However, deep equilibrium models in general are emerging as an important alternative to traditional DNNs, since they perform as well in many domains with reduced memory requirements (e.g. [Bai et al. 2020](https://arxiv.org/abs/2006.08656)). We think that the newness of the monDEQ actually means our paper can serve an important role by highlighting the benefits of these models. Beyond the stability guarantees they have over standard DEQs, we've shown that their Lipschitz constants are simple to bound and control, leading to adversarial robustness and generalization guarantees which are hard to come by for standard networks. We hope that by describing these benefits of monDEQ, the paper will serve to encourage further research in the area.

---

### Author Response · Authors · 2020-11-20
**Response to common reviewer comments**

*We thank the reviewers for their detailed reading of the paper and their insightful comments. We would like to address a few of the common comments here. We have also answered specific reviewer questions in individual replies, and directly fixed several minor issues in the pdf.*

**Empirical lower bounds on Lipschitz constants:**
We have updated the pdf to include empirical lower bounds for the DNNs in Figure 1a (and removed the misleading “naive lower bound” taken from Combettes et al., which is not in fact a bonafide lower bound). All of the lower bounds are computed by taking the max gradient norm over 50k randomly sampled points. In the final version of the paper, we will include empirical lower bounds computed in this way for all other models.

**Test error of monDEQs vs standard NNs:**
We are now reporting the test error of the monDEQs (width=40) used in Figure 1b, which can be found in Figure 6 in the appendix. The DNNs in figure 1a all obtain between 2.8% and 3.2% test error, while the monDEQs show generally increasing error with $m$, from around 2.4% up to 4.2% for $m$=20. In the final version of the paper we will include the accuracies of all additional models. For example, the convolutional monDEQs in Figure 2d all obtain test error below 2% except for the multi-conv monDEQ with m=20, while the CNN results in Figure 2c are taken from Virmaux and Scaman 2018 which doesn’t report error; we will recreate those experiments and report error in the final version of the paper.

**Adversarial robustness:**
To demonstrate that the low Lipschitz constants of the monDEQs lead to better adversarial robustness than similarly performing DNNs, we have now included an experimental section assessing certified and empirical adversarial robustness (Section 5.3). We observed that the monDEQ with m=20 outperforms DNNs both in terms of certified robustness based on the Lipschitz bounds, as well as empirical robustness against PGD attacks.

---

> ### Author Response · Authors · 2020-11-25
> **Further significant updates**
>
> Thanks again to all the reviewers for helping us strengthen our submission! We have made two further additions to the pdf: a) CIFAR-10 experiments (Lipschitz bounds in Section 5.1 and adversarial robustness in Appendix I) and b) comparisons to additional related work towards the end of Section 2.

---

### Decision · Program_Chairs · 2021-01-07
**Final Decision**

**Decision:**

Accept (Poster)

**Comment:**

The paper is interested in the Lipschitz constant estimation of deep equilibrium models. The estimation of this constant provides us the ability to certify classification decisions and understand robustness as well as has important bearings on the generalization ability of a neural network.  Overall a solid theoretical contribution with rigorous theory in a well-written paper.